# Stability of ecologically scaffolded traits during evolutionary transitions in individuality

Guilhem Doulcier [1,2] ✉, Peter Takacs [1,3], Katrin Hammerschmidt [4,6] ✉ & Pierrick Bourrat [1,3,5,6] ✉

Evolutionary transitions in individuality are events in the history of life leading to the emergence of new levels of individuality. Recent studies have described an ecological scaffolding scenario of such transitions focused on the evolutionary consequences of an externally imposed renewing meta-population structure with limited dispersal. One difficulty for such a scenario has been explaining the stability of collective-level traits when scaffolding conditions no longer apply. Here, we show that the stability of scaffolded traits can rely on evolutionary hysteresis: even if the environment is reverted to an ancestral state, collectives do not return to ancestral phenotypes. We describe this phenomenon using a stochastic meta-population model and adaptive dynamics. Further, we show that ecological scaffolding may be limited to Goldilocks zones of the environment. We conjecture that Goldilocks zones—even if they might be rare—could act as initiators of evolutionary transitions and help to explain the near ubiquity of collective-level individuality.

Among the most fundamental and intriguing biological phenomena are cases where collective-level individuals emerge from independently reproducing particle-level individuals. Such evolutionary transitions in individuality (ETIs) appear to be pervasive[1–11]. Although the evolution of multicellularity is typically considered the quintessential example[9,12–14], abiogenesis[15], as well as the emergence of cells[16], organelles through endosymbiosis[17], and eusocial organisations[18], likewise qualify as ETIs. While researchers readily acknowledge that these phenomena have occurred several times during the history of life, there is to date surprisingly little agreement about how the process of transition unfolds or which factors causally initiate and terminate it. Despite the lack of consensus, multiple independent occurrences of similar transitions strongly suggest that there may be general mechanisms promoting ETIs.

Recently, ecological scaffolding has been proposed as a possible scenario for ETIs[19–26]. Key to this scenario is that an ETI can be initiated by a specific kind of externally imposed meta-population structure—a scaffold—consisting of a set of populations of particles confined to bounded, resource-limited patches with limited dispersal (Fig. 1, arrow 1). The patches can differ in their nature, ranging from physical structures to organisms hosting symbionts. In situations where patches can be depleted and regenerated (newly available structures, surfaces, or hosts), the colonisation of new patches and the extinction of existing populations within old patches creates a birth-death process at the collective population level. This collective birth-death process provides an opportunity for natural selection to promote beneficial traits at the collective level (Fig. 1, arrow 2). One benefit of this approach is that it provides solutions to the chicken-and-egg precedence problem for defining new units of selection[19,27]: that is, explaining the emergence of collective-level adaptation without referring to collective-level properties from the get-go. However, the ecological scaffolding scenario is not necessarily the only path to ETIs; it is but one possible mechanism among others[4,28,29].

[1]Philosophy Department, Macquarie University, New South Wales 2109, Australia. [2]Theoretical Biology Department, Max Planck Institute for Evolutionary Biology, Plön, Germany. [3]Department of Philosophy and Charles Perkins Centre, The University of Sydney, New South Wales 2006, Australia. [4]Institute of General Microbiology, Kiel University, Kiel, Germany. [5]ARC Centre of Excellence in Synthetic Biology, Sydney, Australia. [6]These authors contributed equally: Katrin Hammerschmidt, Pierrick Bourrat. ✉e-mail: guilhem.doulcier@normalesup.org; katrinhammerschmidt@googlemail.com; p.bourrat@gmail.com

Another benefit is that ecological scaffolding provides a unifying scenario for a range of research questions in the study of ETIs. It does not constitute a new evolutionary principle but rather a novel way of approaching ETIs that takes into account meta-populations and environmental gradients from evolutionary ecology[30], self-renewing networks of patches from epidemiology[31], the effect of population structure and assortment on the evolution of cooperation studied from multi-level selection and game theory[32,33], and feedback between evolutionary and ecological processes from adaptive dynamics[34]. This is in marked contrast to prevailing explanations of ETIs[2,4,5,28,35], which have traditionally attributed a less prominent role to the combination and interaction of these factors. The ecological scaffolding scenario has been used to successfully explain selection for traits associated with higher collective-level persistence and fertility, such as lower particle growth rate in a resource-limited environment[19], particle-particle interactions that ensure transgenerational stability of collective composition[36], and proto-germ-soma differentiation[11].

While previous articulations of the ecological scaffolding scenario provided a path for the emergence of collectives and traits at that level, the mechanism proposed was insufficient. In nature, collective-level entities that are considered individuals typically do not exhibit an external scaffold. Whatever scaffolds there may once have been, collective-level individuals have done away with them over time, just as a successfully constructed building can stand true after the scaffolding that supported it has been removed. Accordingly, it appears that an ETI is more fully realised when selected collective-level traits remain evolutionarily stable even in the absence of the initiating scaffold (Fig. 1, arrow 3). Proponents of ecological scaffolding have dubbed this phenomenon *endogenisation*[19,21,22,25].

Scaffold endogenisation remains an all-important but unresolved issue for this scenario. Previous explanations have not provided convincing reasons for differentiating cases in which collective-level entities persist as individuals from cases where they would disintegrate in the absence of scaffolding conditions. The standing challenge, then, is to provide an adequate mechanistic explanation of the endogenisation process. We address this challenge by delineating the conditions under which a scaffolded, collective-level trait can remain evolutionarily stable even when the scaffold is lifted. The metaphorical idea of lifting the scaffold can be implemented or operationalised in several ways. While it could be implemented by removing all externally imposed meta-population structures (as in refs. [19,21]), we here assume that an externally imposed meta-population structure is ever-present and instead choose to adjust its defining parameters so that scaffolded traits would not have evolved in the first place (i.e., by reverting them to pre-scaffolding parameter values). This approach enables us to show that the evolution and stability of collective traits in the ecological scaffolding scenario involve an evolutionary hysteresis effect[37,38] mediated by changes in an externally imposed population structure.

In this work, we describe a general mechanistic model (Non-endogenised traits) for a wide range of meta-population structures and derive formal conditions (Formal conditions for trait endogenisation) under which ecological scaffolding can lead to endogenisation once the scaffold is lifted. Using this model, we thus show exactly how a collective-level trait can be endogenised (Endogenised traits). We subsequently explore some consequences of this phenomenon for ETIs. One particularly noteworthy finding is that the specific environmental conditions under which collective individuals emerge might only be a subset of the environments in which collective individuals can thrive. This suggests the existence of "Goldilocks" zones, which could be seen as global initiators of ETIs. New collective traits could originate in these restricted zones and subsequently enable expansion into more remote, initially inaccessible areas of the environment (Goldilocks zones as ETI initiators).

## Results

### Stochastic meta-population model of ecological scaffolding

To explore a wide range of scaffolding conditions, we model a meta-population of a fixed number $D$ of patches connected by the edges of a graph, each initially containing $R$ resources, and particles that each carry a mutable trait $\theta$. Each patch can harbour a set of (one or many) particles, which we refer to as a collective. Particles consume resources at a rate 1 and, upon consuming resources, either duplicate or become a propagule with probability $p$ and $(1 - p)$, respectively (Fig. 2a). When a patch has no resources left, all particles within it die instantly and resources are replenished to the initial value $R$. The probability $p$ depends on both the trait $\theta$ and the local state of the patch

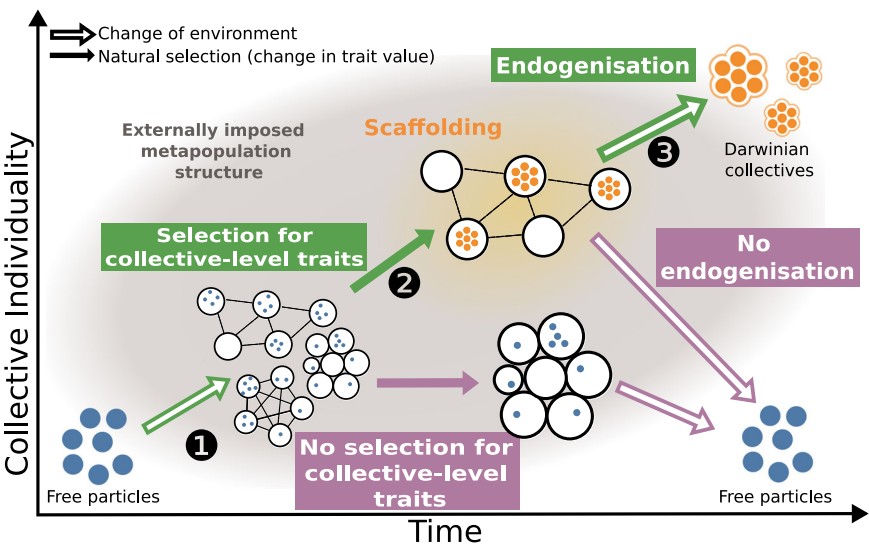

**Fig. 1 | Endogenisation of scaffolded traits.** Scaffolded traits evolve when the population is structured in a specific way. By modifying the meta-population structure (i.e., when the scaffold is removed), the properties can either revert to their ancestral values (no endogenisation) or keep the same values (endogenisation). Under the ecological scaffolding scenario, an evolutionary transition in individuality involves the formation of an externally imposed meta-population structure from free particles (arrow 1). Trait scaffolding is the selection of a trait value in specific conditions of the environment (i.e., scaffolding environment, arrow 2) that promotes survival and reproduction at the collective level. A scaffolded trait is said to be endogenised if it remains stable even when the meta-population conditions that were required for its emergence are perturbed (arrow 3).

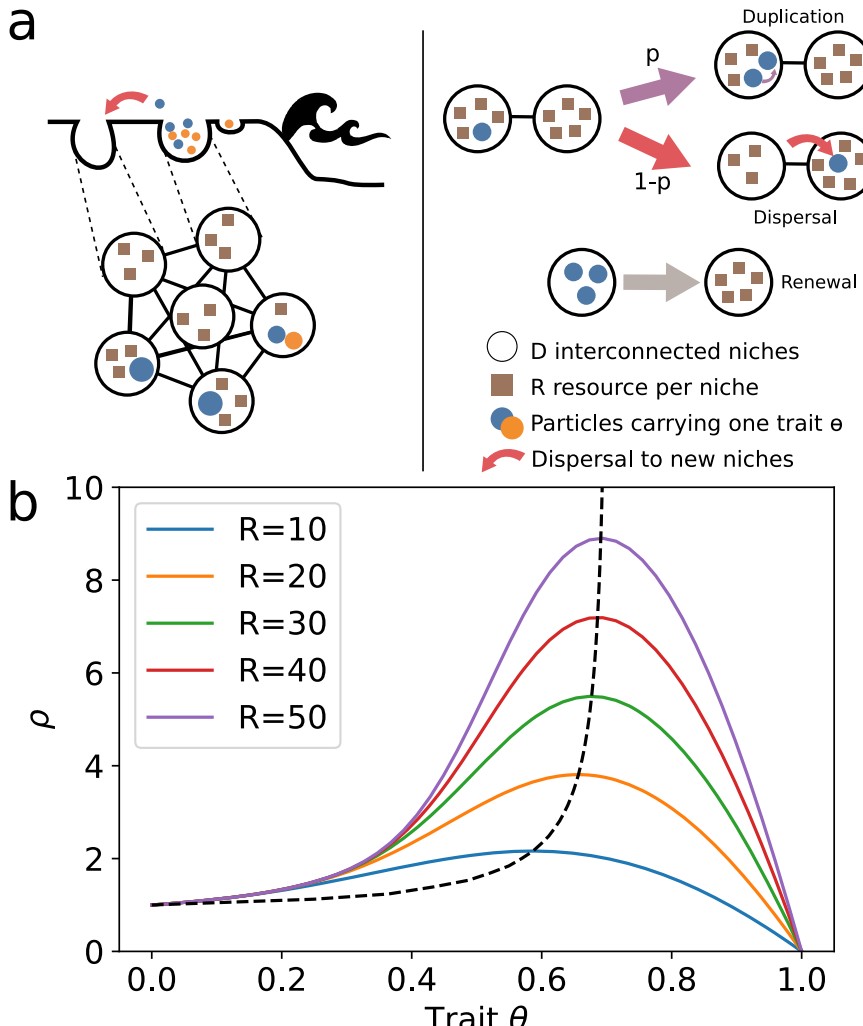

**Fig. 2 | Evolutionary dynamics. a** Meta-population model: in our model, there are D available patches that initially contain R resources. Particles reproduce or migrate with rate 1. Particles carry one trait $\theta$ that determines the probability that the particle will reproduce ($p(\theta)$) or migrate to an empty connected patch ($1 - p(\theta)$). When all resources are consumed, all particles within the patch die and the resources are replenished to the initial value R. **b** Intrinsic duplication-dispersal ratio: when the probability of dispersal or duplication is intrinsic and constant ($p(\theta) = \theta$), there is a global optimal trait value ($\theta^*$), which is an evolutionarily stable strategy, for each resource level R. The dashed line represents the ($\theta^*, \rho(\theta)^*$) values for R in [2, 100]. If the meta-population changes, the trait value tracks the new optimum, but there is no endogenisation.

(predominantly, the number of particles within it). We study two functional forms for the dependency of $p$ on $\theta$ (which are detailed in Non-endogenised traits and Endogenised traits). When a particle migrates from a patch, it colonises an empty connected patch. If no such patch exists, the propagule particle dies. These rules constitute a Markov jump process, with events detailed more formally in Supplementary Note 1. Provided that the graph encoding the adjacency of patches is complete (i.e., that particles can migrate from any patch to any other empty patch), the process can be approximated by a simple set of two ordinary differential equations (ODEs), as detailed in Supplementary Note 2. The simplified ODE system has two key parameters for a given trait $\theta$ value: the lifetime of a collective $\tau(\theta)$ and the lifetime expected number of propagules for a collective $\rho(\theta)$. These two values can be derived from the stochastic system, as shown in Supplementary Note 3. The evolution through natural selection of the trait $\theta$ in the simplified ODE system is described with the framework of adaptive dynamics[34]. The derivation of the invasion fitness gradient is presented in Supplementary Note 2.

Within this general scenario, we can study the problem of endogenisation. Endogenisation of a trait occurs when the scaffolded trait value persists after the scaffolding conditions revert to pre-scaffolding

conditions[19,21]. First, we demonstrate that a simple trait value (i.e., taking a constant intrinsic value for the duplication-to-dispersal value, $p = \theta$) can change as the scaffolded meta-population reverts to a non-scaffolding state (Non-endogenised traits). This shows that scaffolded traits are not necessarily endogenised (i.e., there can be instability when the environment is reverted to pre-scaffolding conditions). We thus establish formal conditions for the endogenisation of traits based on the structure of the trait-fitness landscape (Formal conditions for trait endogenisation). We then choose a trait fulfilling these formal conditions (i.e., taking a duplication-to-dispersal ratio that is sensitive to local population density) and show that its value is stable even if the scaffolding conditions are lifted (Endogenised traits). Finally, given a gradient in externally imposed meta-population structure, we show how collective-level entities can colonise areas beyond the limited zones of the environment that present scaffolding conditions. These zones can effectively become initiators of ETIs through a phenomenon we call the "Goldilocks Zone Effect" (see Goldilocks zones as ETI initiators).

## Non-endogenised traits
Here, we examine the simplest functional form for the trait and study its evolutionary response to changes in the meta-population

parameter. Consider a case where the mutable trait $\theta$ is carried by each particle and directly encodes the rate at which cells duplicate ($p$) or leave the patch ($1 - p$), such that $p(\theta) = \theta$, independently of their surroundings. This trait encodes the investment towards particle duplication or dispersal in the shape of a linear tradeoff. At the collective level, this translates into a nonlinear tradeoff between the life span of collectives $\tau(\theta)$ and the average number of propagules they produce $\rho(\theta)$ (due to demographic effects, as illustrated in Supplementary Fig. 2). Note that, in this model, the collective-level fertility or survival is not the average or the mere sum of particle-level traits but rather a nonlinear function of the trait (this feature sets our model apart from other models of ETIs that statistically derive the values of some collective traits from the sum or average of particle-level traits[39,40]). The limited resources in the patches are either allocated to producing more particles or to producing propagules. The behaviour of collectives for different values of $\theta$ can now be detailed.

The population is not viable for the two extreme trait values, $\theta = 0$ and $\theta = 1$. To see why, consider the ecological dynamics of a unique patch seeded with one cell. If $\theta = 0$, particles never reproduce and always move from one patch to the next. In this limiting case, the lifetime reproductive output of a patch is always $\rho(0) = 1$ and its life span is, on average, $\tau(0) = 1$. Neither the population of particles nor that of collectives ever grows in size because no particle reproduction occurs. Conversely, if $\theta = 1$, particles never become propagules. They duplicate until all resources are expended and the local population goes extinct. Thus, $\rho(1) = 0$. Intermediate values of $\theta$ have higher values of $\rho$, as shown in Fig. 2b.

Now, if we consider the ecological dynamics of the entire metapopulation, we know from the analysis of the ODE system that in the case of fully connected patches, the population is only viable if $\rho(\theta) > 1$ and reaches an equilibrium occupancy that depends on the value of $\rho$ (see Supplementary Fig. 4). This tells us that the population of collectives can only maintain itself if the expected number of propagules during its lifetime is higher than 1. This condition on the collective-level trait $\rho$ translates to particle-level traits: the population is viable for values of $\theta$ strictly higher than zero and smaller than a threshold $0 < \theta_{max} < 1$, such that $\rho(\theta_{max}) = 1$.

To predict the evolutionary trajectories of the system, we turn to invasion analysis. Consider a homogeneous population at ecological equilibrium. The fate of a single new mutant particle with a different value of $\theta$ in a new patch can be predicted using the ODE model (Supplementary Note 2). It has either a positive growth rate (black zone in the pairwise invasibility plot, Supplementary Fig. 5), and thus a possibility to become fixed in the meta-population, or a negative growth rate (white zone, Supplementary Fig. 5). The value of the invasion fitness results in an evolutionarily stable strategy (ESS; $\theta^*$) that is convergence-stable (i.e., a resident population with a trait value that is close but not exactly equal to the ESS can be invaded by mutants with a trait value that is even closer to the ESS) and evolutionarily stable (i.e., a resident population with the trait value $\theta^*$ cannot be invaded by mutants with a close trait value). The value of the ESS depends on the number of resources $R$ in a single patch.

Finally, consider the case in which $\theta$ can mutate freely, starting from an ancestral value $\theta_0$ that is viable ($0 < \theta_0 < \theta_{max}$). When mutations are so rare that populations reach a new ecological equilibrium between the emergence of new mutants (i.e., the ecological and evolutionary time scales are separated), adaptive dynamics[34,41] predicts that the meta-population will converge towards the ESS. Stochastic trajectories of the system are accurately predicted in this way, as shown in Supplementary Fig. 3a, b. The value of the ESS $\theta^*$ increases with higher values of $R$, meaning that an increased rate of duplication is favoured when resources are plentiful (Fig. 2b, dashed line, and Supplementary Fig. 6).

In this model, the selected trait value depends on the meta-population parameters (here, the meta-population is fully connected;

the only parameter is the number of resources in a patch). If the parameter $R$ changes, the selected trait value simply tracks the optimal probability associated with the change, which means there is no endogenisation.

## Formal conditions for trait endogenisation

Having shown that scaffolding does not entail endogenisation, we now establish the conditions for endogenisation in this model. A formal derivation of these conditions can be found in Supplementary Note 4.

Consider a two-dimensional landscape that maps the state of the environment and the value of a collective trait to its fitness in this environment. This simplified landscape (illustrated in Fig. 3) allows us to describe the conditions where scaffolding and endogenisation are possible. To keep the model simple, we consider only two states of the environment—"scaffolding" or "non-scaffolding"—as well as two collective trait values—"no collective organisation" or "collective organisation." This results in four trait-environment combinations. By "collective organisation," we mean that the trait exhibits behaviour different from the intrinsic duplication-dispersal trait, or what we refer to as "non-aggregativity" (see Discussion).

In the initial condition, we assume that the environment is non-scaffolding and there is no collective organisation. This state is evolutionarily stable. A scaffolding experiment is then conducted. At time 0, the environment is changed to a scaffolding condition (Step 1), held constant for a duration $T$ (Step 2), and then reverted to the non-scaffolding condition (Step 3).

To claim that an environment is scaffolded here implies that a change in population structure eventuates in the evolution by natural selection of a collective-level trait (Steps 1–2). A collective trait is said to be endogenised if it is evolutionarily stable after the scaffolding conditions are lifted (Step 3). Endogenisation could be figuratively depicted as a situation in which the population retains a memory of its scaffold. However, the sense of retention thereby implied must be understood as stronger than that which might be associated with merely encountering and evolving in the wake of some historically contingent set of circumstances. Less metaphorically, endogenistation is an instance of a phenomenon called "hysteresis" in dynamical systems theory, which is ubiquitous across all domains of biology[37,38].

Scaffolding followed by endogenisation (Steps 1–3) is only possible if three conditions are fulfilled. First, both trait values must be evolutionary stable in the non-scaffolding environment. Second, the no-collective-organisation trait value must be evolutionarily unstable in the scaffolding environment. Third, the trait value of the population after a duration $T$ must fall within the basin of attraction for the collective-organisation trait in a non-scaffolding environment. In other words, the transition to a scaffolding environment from a non-scaffolding environment and then back again allows populations to circumvent valley in the fitness landscape that exists only in the non-scaffolding environment and would prevent evolution towards collective organisation if no scaffold was imposed.

These conditions are not fulfilled in the simple model without the particle-particle interactions presented in the Non-endogenised traits section. Indeed, there is a single ESS for each value of $R$. Consequently, there is no fitness valley in the non-scaffolding environment to be circumvented. Some interactions between particles are necessary for successful endogenisation in this model.

In the next two sections, we follow a trait that fulfils the endogenisation conditions. In Endogenised traits, we study the evolutionary stability of the derived trait value in the face of environmental change. In Goldilocks zones as ETI initiators, we explore the consequences of this stability when the population is put in an environmental gradient.

## Endogenised traits

In this section, we present a variation of the model in which the probability for a particle to either duplicate or disperse depends on its

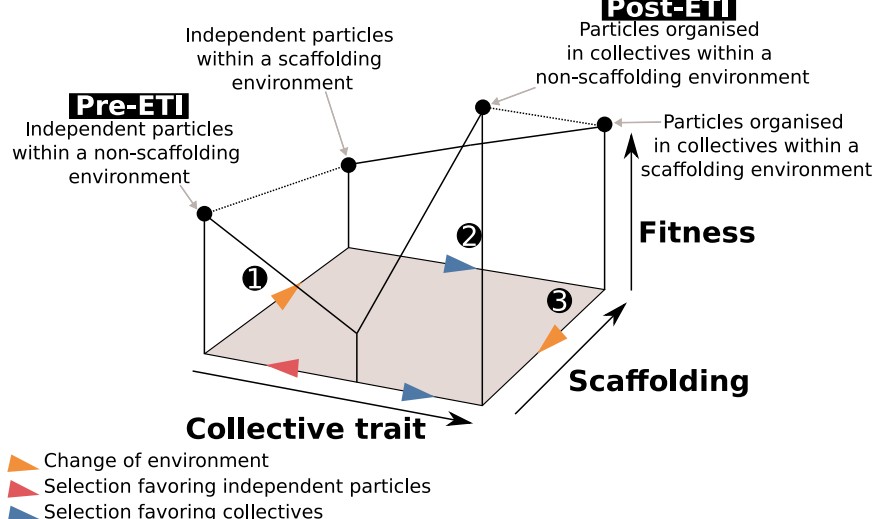

**Fig. 3 | Conditions for endogenisation.** A simplified scaffolding scenario in three steps: **1.** Change in the environment **2.** Selection of the collective trait in the scaffolded environment (Scaffolding) **3.** Change in the environment. A collective trait is said to be scaffolded if a change in the population structure results in its evolution by natural selection (Steps 1–2). A collective trait is said to be endogenised if it is evolutionarily stable after the scaffolding conditions are lifted (Step 3). ETI = evolutionary transition in individuality.

ability to sense the presence of other particles in the patch (Fig. 4). The model with particle-particle interactions fulfils the endogenisation conditions outlined in the previous section. This trait model accordingly provides a simple way of depicting the effect of a control mechanism (receptors and regulation), one which has the capacity to change the probability for a given particle to disperse or duplicate. If, for instance, the particles in question are cells, this could represent the effect of direct cell-cell signalling or that of a quorum-sensing molecule that results in activating or repressing physiological pathways. If, in contrast, the particles in question are self-replicating molecules, non-covalent interactions could lead to a change in the probability of dispersal, depending on the density.

Within this model, the highest number of propagules a collective can produce is $\rho = R/2$. This assumes that a particle always duplicates if it is alone and always disperses if that is not the case. We refer to this phenotype as "perfect particle coordination." In this case, at any point in time, collectives are relatively small (between one and two particles), even though they produce many particles and exploit the resources in the patch optimally during their lifetime. This stark selection for the smallest viable collective size possible results from simplifying assumptions of our model that make it amenable to analysis. The simplifications include a perfect response mechanism for particle coordination (i.e, perfect sensory information, deterministic behaviour, and no delay). As a consequence, there is no competitive advantage for collectives to be composed of more than two particles at a given time (i.e. bigger investment in soma does not translate into an increase in viability). We show in Supplementary Note 5) that these assumptions can be relaxed to reach bigger collective sizes (with an average maximal size around 4 particles) at any point in time. However, the hysteresis behaviour that interests us here tends to disappear when the number of particles becomes large.

Two other simplifying assumptions of our model limit the advantage of large population size within patches. The first is the absence of particle death within patches. This reduces any advantage associated with preventing extinction by having many particles in the patch. The second assumption is that propagules can only colonise empty patches. Although several phenotypes can compete within a single patch, there is a bottleneck of one particle when patches are newly colonised.

Now we describe the model for the evolution of coordination between particles. Let the mutable trait $\theta$ quantitatively encode the coordination between particles (see Fig. 4a) in the following way:

$$p(\theta, n) = \begin{cases} \theta & \text{if } n = 1, \\ 1 - \theta & \text{if } n > 1. \end{cases} \quad (1)$$

If $\theta = 1$, the coordination is perfect, and the particle always duplicates if it is alone ($p(1,1) = 1$) and disperses otherwise ($n > 1, p(1, n) = 0$) (see Fig. 4b). If $\theta = 0.5$, particles act independently of one another and the probability for a particle to duplicate or disperse is the same regardless of the state of the patch (see Fig. 4b). Note that if $\theta = 0.5$, this model and the simple model introduced in the previous section are equivalent. If $\theta = 0$, particles are anti-coordinated: they always disperse if they are alone ($p(0,1) = 0$) and reproduce otherwise ($n > 1, p(0, n) = 1$). This translates to a strategy where particles always disperse to new patches without ever duplicating. Finally, other values of $\theta$ correspond to different degrees of coordination between particles.

As before, we again consider just two environments: non-scaffolding and scaffolding. The non-scaffolding environment is composed of large patches ($R = 100$), whereas the scaffolding environment is composed of smaller patches ($R = 20$). As shown in Fig. 4c, this setup meets the three conditions defined in the section Formal conditions for trait endogenisation. It presents two alternative evolutionarily stable states in the non-scaffolding environment (slightly anti-coordinated particles $\theta \approx 0.41$ and perfectly coordinated particles $\theta = 1$) separated from each other by a fitness valley (with a minimum at $\theta \approx 0.66$). However, in the scaffolded environment, only the coordinated phenotype ($\theta = 1$) is evolutionarily stable. In the following, we assume that the ancestral state is $\theta_0 = 0.5$ (independent particles) in a non-scaffolding environment and let the system stabilise to the slightly anti-coordinated stable state ($\theta \approx 0.41$). We then ask whether better coordination (higher values of $\theta$) can evolve due to scaffolding (i.e. due to reducing $R$ to 20) and subsequently be endogenised in the sense of the Formal conditions for trait endogenisation section (i.e., be stable even if $R$ is reverted to 100).

The duration of the scaffolding must be sufficiently long to allow crossing of the fitness valley. Given the setup just described, we begin by allowing an initial population of particles with ancestral trait $\theta_0$ to

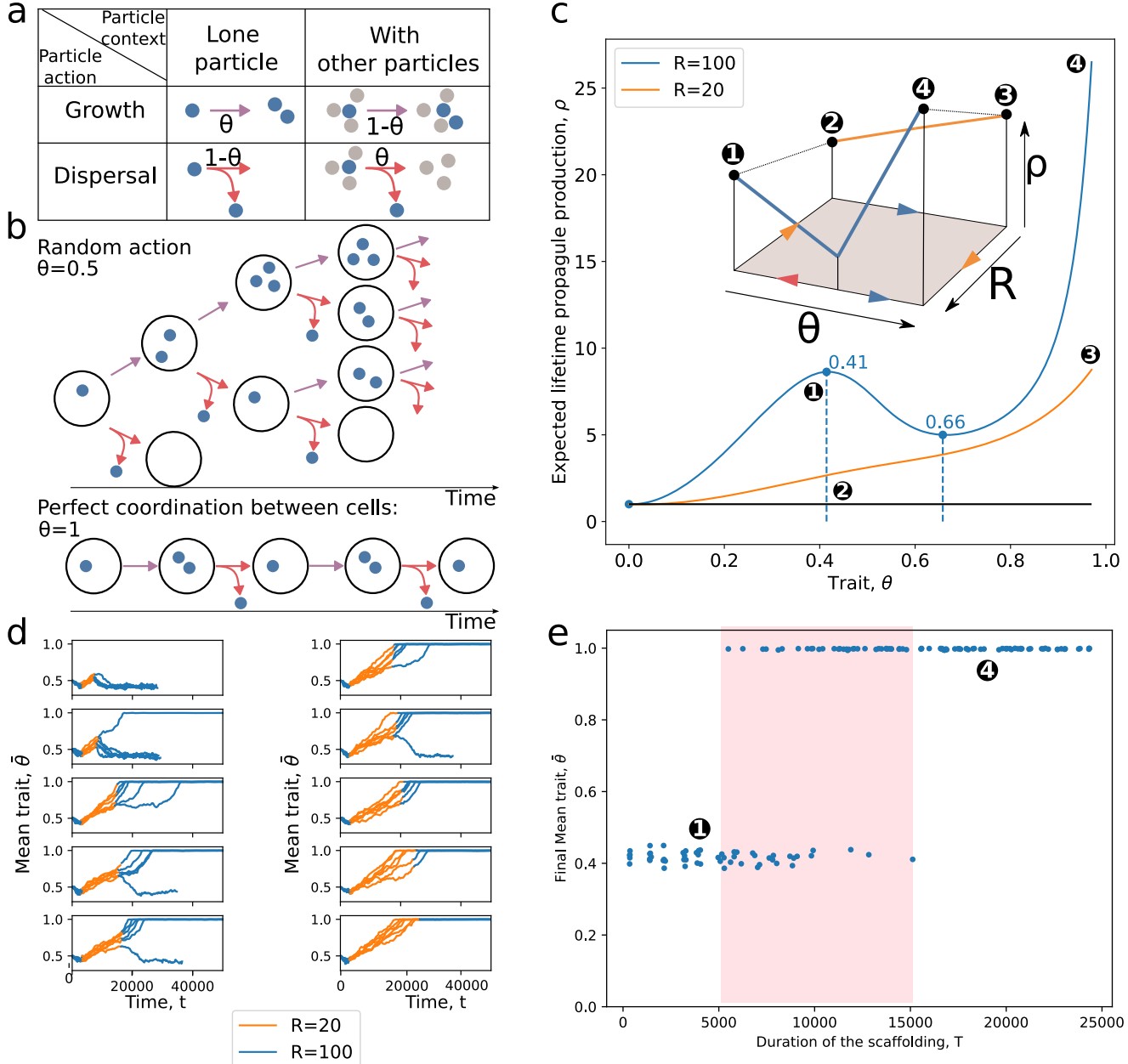

**Fig. 4 | Endogenisation of a scaffolded trait value. a** Model. In the density-dependant duplication-dispersal ratio, the trait value $\theta$ controls the probability to duplicate rather than disperse when the particle is alone in the patch and, conversely, to disperse rather than duplicate when the particle is not alone. If $\theta = 0.5$ (ancestral trait value), particles act the same way regardless of the local density of particles, as in the intrinsic ratio model (Fig. 2) with value 0.5. **b** Extreme traits values. The optimal value of $\theta$ is 1 because it produces the most propagules without wasting any resources (e.g., by leaving a patch before all resources are expended). **c** Propagule production as a function of the trait value. Patches contain a quantity $R$ of resources. In larger patches ($R > 39$, see Supplementary Fig. 7), there is a fitness valley between the ancestral value 0.5 and the coordinated value 1. This valley does not exist in small patches. Hence, the trait value fulfils the conditions for endogenisation. **d** Evolutionary trajectories. If particles from large patches are put into smaller patches for a limited duration (orange part of the trajectory), the average trait value tends towards 1 as higher values are selected. However, if they are put back in larger patches too soon, they revert to the ancestral value. **e** Evolutionary endpoint as a function of the duration of scaffolding by small patches. Intermediary scaffolding durations (red area) do not always lead to endogenisation due to stochastic effects.

evolve in the non-scaffolding environment for a duration $T_0$. The value of the environment is then changed so that it satisfies the conditions for scaffolding over a duration $T$. Finally, we revert the environment back to the non-scaffolding value.

Figure 4d shows simulations of such temporary scaffolding experiments (see also Supplementary Fig. 3c, d for simulations in constant environment). Initially (before $T_0$), the populations evolve towards a mean trait value of $\theta < \theta_0$. When the scaffold is applied, the sole evolutionarily stable trait value becomes $\theta = 1$. If the scaffold is lifted after the basin of attraction around $\theta = 1$ has been reached, the mean trait in the population is stable and endogenisation has occurred. However, if the scaffold is lifted before the basin of attraction around $\theta = 1$ has been reached, the population mean trait could revert to $\theta \approx 0.41$, depending on whether there were enough individuals in the population with a value of $\theta$ low enough to have crossed the fitness valley at $\theta \approx 0.66$.

Figure 4e shows the proportion of successful scaffolding for different values of $T$. Note that intermediate values of $T$ do not always

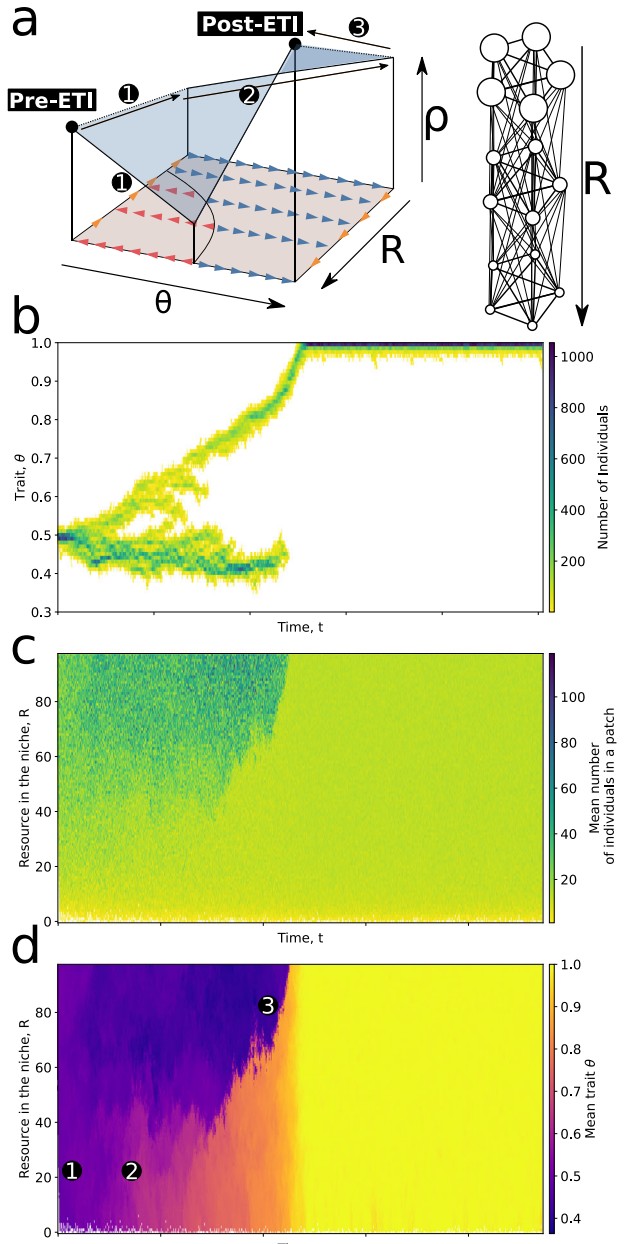

**Fig. 5 | Goldilocks zone. a** Patch network. In this new model, the patches are structured along a gradient from resource-rich patches ($R = 50$) to resource-poor patches ($R = 0$). All patches from a single layer are connected to one another, as well as to patches of the richer and poorer layers. **b** Trait distribution through time. The initial trait distribution is centred around $\theta = 0.5$. **c** Number of individuals in each category of patches through time. Initially, only the richest patches ($R = 50$) contain particles. These subsequently migrate down the gradient (1). **d** Mean trait value in each category of patches through time. Note how higher values of $\theta$ first emerge in resource-poor patches (2) and then migrate back to resource-rich patches (3). ETI = evolutionary transition in individuality.

result in endogenisation. This non-deterministic behaviour comes from the stochastic nature of the mutations that do not occur exactly at the same moment in replications of the simulation.

Overall, the density-dependent particle dispersal-duplication ratio studied in this section fulfils the three conditions for endogenisation and accordingly displays a corresponding evolutionary trajectory: when patch size is reduced, coordination between particles is selected. Coordination is retained (i.e., is evolutionarily stable) even when patch sizes return to their ancestral values.

## Goldilocks zones as ETI initiators

In this section, we replace the temporary scaffold (changing the environment for a limited duration) with a standing environmental gradient that features both scaffolding and non-scaffolding conditions. Consider a diversity of patches along a size gradient from very large patches ($R = 100$) to very small patches ($R = 2$, see Fig. 5a). There are 10 patches for each resource richness class and 98 different classes, for a total of $D = 980$ patches. Dispersal is only allowed between patches of the same or adjacent richness (e.g., particles in a patch $R = 50$ can disperse towards patches with $R = 50$, $R = 49$, or $R = 51$). Initially, ancestral particles with $\theta_0 = 0.5$ are seeded in the richest patches ($R = 100$) and the dynamic of the population is simulated. Fig. 5b−d shows the result of such a simulation.

The trajectory can be summarised by noting three qualitatively distinct phases. In the first phase, the population expands to occupy all the patches that are viable (this happens very fast on the evolutionary timescale and is not visible in Fig. 5b−d). However, the patches that are too poor in resources are not populated most of the time because the populations are too small (we call this lower bound $R_0$ hereafter). During the second phase, the populations evolve towards the optimal trait value in each part of the gradient. Two qualitative behaviours exist along the gradient (separated by a fold bifurcation in $R^* = 39$, as shown in Supplementary Fig. 7). For patches where $R > R^*$, the populations evolve towards the non-scaffolded peak value $\theta \approx 0.41$, while for $R < R^*$, they evolve towards the scaffolded value $\theta = 1$. Finally, particles that have evolved the scaffolded phenotype for coordinated dispersal and reproduction eventually disperse back into the rich patches and outcompete the ancestral type.

The area in the gradient between the viability limit and the scaffolding limit ($R$ such that $R_0 < R < R^*$) therefore constitutes a "Goldilocks zone," where the population structure is suitable for ecological scaffolding. As in the children's tale where only one of the three bears' belongings is just right for Goldilocks, only some of the available meta-population structures will be just right for scaffolding. The Goldilocks zone thus acts as a population source (in the ecological source-sink sense) for coordinated collectives. The ecological range of coordinated collectives (i.e., the portion of the gradient in which their population is stable) is wider than the possibly small Goldilocks zone, which is the only place they can initially evolve. More generally, this suggests that a Goldilocks zone can act as an initiator of the ETI.

## Discussion

The present article investigates whether an ETI can be initiated by an externally imposed population structure (the scaffold) and describes the conditions under which newly selected properties could persist even if the environment that promoted their emergence has changed. We have shown that this outcome can be the result of a hysteresis effect: changing the parameters of the meta-population structure changes the nature of the evolutionary landscape, making stable but initially unreachable traits accessible. This was demonstrated by way of two distinct scenarios, one temporally scaffolded and the other spatially scaffolded.

The concept of individuality has been extensively debated in the philosophical and biological literatures[10,42–48], with different criteria for defining what constitutes an evolutionary individual. However, three commonalities regarding conditions for individuality (at a given level) can be identified: (1) the existence of boundaries separating the collective from its environment or discretisation of collectives[21,45,49–51], (2) the existence of what can be regarded as genuine collective traits that legitimise the notion of collective individuals[4,40], and (3) the stability of collective structures[52,53]. We now discuss the ecological scaffolding scenario presented here in light of these three requirements. In particular, we discuss how each of these conditions arises within the scaffolding-endogenisation scenario for ETIs.

The emergence of discrete collectives corresponds to the problem of collective formation for ETIs[2,3]. The existence of population structure in a population of particles can produce discrete collectives. However, it occasionally produces mere neighbourhoods. It has been argued that the latter do not in fact represent genuine collective-level units or individuals[49,50]. In response to the inadequacy of bare population structure as a definitive criterion, several "individuation mechanisms" for discrete collectives have been proposed[45,54]. These can be categorised according to two general ways by which particles can assemble into discrete collectives, following the "coming together/staying together" distinction proposed in[55].

First, particles "staying together" and cluster fragmentation can give rise to discrete collectives[56]. This can occur in nascent multicellular systems with imperfect cell division, such as snowflake yeasts[28,57], algal clusters[58], and cyanobacterial filaments[59]. Second, discrete collectives can result from the aggregation of particles "coming together." For instance, in aggregative multicellularity (e.g., aggregative amoebae), cell-cell adhesion can create discrete clusters, potentially sorting cells through differential attachment[60–63]. Third, fusion of particles can lead to discrete entities. This is particularly visible in ETIs arising via endosymbiosis, where the engulfment of one particle type provides the template for a discrete population structure[17]. Finally, particle assortment may be the result of limited dispersal in a spatial environment[32,64]. However, if this leads to assortment in continuous populations, it will not yield genuinely discrete clusters. One way to get discrete clusters in a continuous environment from limited dispersal and signalling alone (i.e., without particle attachment) is to involve Turing patterns[65], but these require a minimal complexity of particles that necessarily assumes two kinds of interaction. Overall, both spatial self-organisation and compartmentalisation have been shown to lead to multi-level natural selection[66].

In contrast, the ecological scaffolding scenario establishes discretisation via an externally imposed meta-population structure that initiates an ETI[19,20,24]. The core idea is that the meta-population structure is externally imposed and, thereby, independent of the nature of the particles. This externally imposed structure nonetheless changes the shape of the fitness landscape (making it a changing fitness "seascape," in the sense of[67]). However, there is no need to presuppose complex behaviours like preferential or differential attachment, signalling, or imperfect division, as in other models. Although our model does not feature particle-particle attachment, it is noteworthy that this type of attachment is not a prerequisite for an ETI. For instance, in some contexts, coordination between cells on patchy resources could play the same role as particle attachment[68] or eventually promote it[69]. At least in terms of general applicability, the fact that our model makes only minimal assumptions about the nature of the particles it models is a strength. A direct benefit is that it can be used to model different kinds of ETIs, ranging from those that might involve molecules and a scaffold consisting of iron monosulphide precipitates[16] or dividing supramolecular vesicles[70,71] (as in models of the origin of the first cell) to ETIs that involve cells and physical support (as in models of early multicellularity)[72], and even ETIs involving organisms and patchily distributed resources (as in models of eusociality)[73,74].

The advantage of minimising assumptions about the nature of particles is of course most obvious when attempting to study early ETIs involving simpler particles or the commonalities that may exist between ETIs. In addition, our model could also be extended to cases where the scaffold is not entirely external but instead arises from the activities of particles themselves. For instance, we could suppose that a patch represents the host and the particles represent endosymbionts, following the endosymbiosis hypothesis for some ETIs. The scaffold, in such a case, would be represented by one of the two partners. The parameters of our model could then be amended to align the replenishment of resources or the creation of new viable patches with the life cycle of the host, which only further demonstrates the versatility of the ecological scaffolding scenario.

From a technical point of view, our model improves on other models for the ecological scaffolding scenario for ETIs[19,20] by allowing overlapping collective generations and introducing more complex spatial structures (i.e., by using a graph of patches rather than an unstructured island model with non-overlapping generations[19] or a continuous resource field, as is the case in a Cellular Potts model[63]). Future work should further explore the parameterisation of the scaffold and create an even more ecologically realistic model. The effect of structuring a population in patches with eco-evolutionary dynamics has been studied in meta-populations[30] and in stepping stone models[75]. However, our model is different in that the migration of particles towards patches that are already occupied is considered negligible. Such mixing could have non-trivial effects on the topology of the fitness landscape, making both scaffolding and endogenisation more difficult (i.e., if the absence of migratory mixing leads to our model and high migratory mixing leads to an unstructured population model, it would be natural to investigate the effects of limited mixing in future work).

Moving on to the second requirement for collective-level individuality, a central question in the study of ETIs beyond the constitution of discrete collectives is whether the evolutionary dynamics being observed result in genuine collectives or merely fortuitous group benefits for independent particles. A (sometimes heated) discussion along this line was initiated by Williams[8,76–79]. One widespread approach to this problem hinges on a comparison of collective-level traits against particle-level traits. Any collective-level trait is necessarily a function of the particle-level traits carried by its constituents (unless we accept a form of strong emergence). However, this function might be complex. In practice, a discrepancy between the value of a collective-level trait and the value taken by a linear function (e.g., averages or sums) over the trait values that constituent particles would exhibit if they were solitary can provide a way to operationalise (weak) emergence and serve to identify a collective level that is merely a "byproduct" of the lower level[8]. In the literature, this strategy can be framed in terms of trait (non)-aggregativity[50,51,80], counterfactual approaches to fitness[81,82], indirect genetic effects as per quantitative genetics (refs. [83],[84] chap. 22), and multi-level selection (MLS1-MLS2)[8,77,85].

The evolutionary trajectory presented in this manuscript features an increase in individuality at the collective level, following the foregoing distinction between a collective-level byproduct and a genuine collective trait. Two situations are considered for comparison. In the first case (intrinsic particle dispersal-duplication ratio), the probability of duplicating or migrating is independent of external cues to the particle; a collection of particles only displays an aggregation of identical behaviour. In the second case (density-dependent dispersal-duplication ratio), this probability depends on the coordination between cells. Their behaviour is not reduced directly to the average over individual behaviour, as it would be if they were alone.

Thus, the evolutionary dynamics described here begin with an ancestral state containing uncoordinated particles on a patch (a mere collective-level byproduct) that is replaced by a derived state where particle behaviour is coordinated and collectives are composed of coordinated particles. This derived state resides further along the evolutionary trajectory towards collective-level individuality. Coordination, in addition to being a form of interaction, can also be regarded as a form of functional integration between the particles, a feature that has been associated with a higher degree of individuality[10,47,86–88].

The stability of (nascent) collectives is the final piece of the puzzle. This topic of group maintenance complements group formation[2]. The stability of early multicellular organisms in the face of single-cell revertants has been identified as a major threat to nascent higher-level entities[7,53,89,90]. Mechanisms that can explain the stability of new

collectives have been studied for various ETIs. For instance, the existence of "ratcheting" mutations that reduce the fitness of particles when they are isolated has been proposed for the evolution of multicellularity[52]. More generally, stability may also come from the fact that collectives can overcome constraints that bear upon independent particles, by allowing collective-bound particles to either outcompete them directly or subsist in previously unattainable environments ("tradeoff-breaking", see ref. 91).

The problem of stability for nascent collectives in the ecological scaffolding scenario can be revealed by a simple question: if collective-level traits are selected as a consequence of a change in the environment (i.e, the scaffolding), can those traits remain stable when that change is reverted? We argue that the stability of a trait selected by ecological scaffolding is a problem of hysteresis. Hysteresis phenomena occur when dynamical systems display a kind of memory of their past states; they are ubiquitous in all domains of biology (reviewed in ref. 38). Hysteresis loops arise in systems with alternate stable states, where an external force pushes the system towards a different stable state, but reverting the external force does not lead to reversion of the state. Here, the alternate stable states are ESS corresponding to the different levels of individuality, and the external force is the scaffolding that reduces the amount of resources in a patch.

Our results delineate three necessary conditions for endogenisation: (i) the presence of a fitness valley (ii) that does not exist in the scaffolding environment and (iii) for which the evolutionary endpoint of the scaffolding process must reside on the other side of the fitness valley. When these three conditions are met, scaffolding allows collectives to circumvent what would otherwise be an uncrossable fitness valley and subsequently stabilise the evolved collective-level trait. For illustrative purposes, we have chosen to use a simple picture that combines a one-dimensional environment with a one-dimensional trait value. In principle, however, there is no limit to the number of traits or the number of environmental parameters that could be tracked simultaneously.

These results can help to develop intuitions about experimental systems. Imagine an experiment in which an ancestral population is exposed to candidate scaffolding conditions for a certain period of time. If no evolutionary response in the collective trait value is observed, new meta-population parameters should be tested. If a reversal of the collective trait is observed upon return to the ancestral environment, this could be because the duration of scaffolding was not sufficient to enable crossing of the fitness valley (this could potentially be resolved by maintaining the scaffolding regime for longer), or possibly because there was no fitness valley at all (this could be tested by looking for qualitative changes in the evolutionary response when trying many different meta-population parameters; see Supplementary Fig. 3).

The ecological scaffolding scenario has been successfully used to interpret evolutionary experiments where the environment is fully controlled by the experimenter[11]. Understanding what it can and cannot do, as well as the effect(s) of meta-population structure parameters, is of empirical interest for the fundamental study of ETIs and for applications involving the artificial selection of communities[20,92–95]. In particular, the stability of selected traits beyond the environments that promoted their initial emergence is a highly desirable property for engineered collectives[96]. Three increasingly strong notions of trait stability might be of interest in this context. First, a trait value might be ecologically unstable, disappearing in one or few generations or being replaced by existing variants in the population. Alternatively, a trait value can be evolutionarily unstable (as in ref. 20), meaning that the trait would disappear in the long run due to the successive invasion of new mutants. Finally, a trait value might be evolutionarily stable (e.g., as a result of endogenisation), meaning that mutants carrying small mutations of this value would not replace the resident population. The last case is the most desirable in applied settings because invading phenotypes that do not contribute to the trait of interest pose a

serious threat to synthetic consortia. Although beyond the scope of the present article, this raises the intriguing prospect that the ecological scaffolding scenario may also apply more generally to non-ETI trait evolution.

A key outcome of our work is that not all externally imposed meta-population structures lead to the scaffolding and endogenisation of collective-level traits. A direct implication is the possibility of zones that can be considered causally operative factors or "ETI initiators," which, for obvious reasons, we have chosen to call "Goldilocks zones." Indeed, it is possible that along an environmental gradient, only some areas meet the conditions for the processes of scaffolding and endogenisation (in the Goldilocks tale, this would correspond to the one bear's belongings that enable the right set of affordances for their trespasser's prolonged stay). When this happens, pre-ETI particles can immigrate into the Goldilocks zone, be selected for collective organisation, and then emigrate out of the zone into another environment. The case of the origins of life (the first ETI chronologically[15]) offers a fitting example. Most working hypotheses about abiogenesis refer to a privileged area of the environment that is uniquely suited to the emergence of collectives. By way of example, it has been hypothesised that the origin of the first cells relied on the iron-sulfide crystalline structure in hydrothermal vents[16,97,98]. The possibility that some variant might evolve in resource-poor conditions and then back-invade regions of a gradient where it was not able to evolve is not a phenomenon limited to ETIs and ecologically scaffolded populations; this can also happen in a non-patch-structured, spatially continuous population[99]. That particular iron-sulfide crystalline structure can act as a Goldilocks zone and could thus have been at the source of the cells that have since colonised our entire planet. Endogenisation of traits would be a prerequisite for this sequence of events. Two additional effects to consider in the environmental gradient scenario (section Goldilocks zones as ETI initiators) compared to the limited-duration scaffolding scenario (section Formal conditions for trait endogenisation) are (1) the direct competition between the derived type by the ancestral type (and whether new collectives can back-invade the ancestral environment) and (2) that newly formed collectives may colonise parts of the environment where ancestral particles were not viable. One of these two phenomena must occur to ensure the long-term evolutionary success of the collectives. This scenario provides us with an especially intriguing consequence: Goldilocks zones can apparently be relatively rare in the overall environment without thereby foreclosing or even limiting the possibility of ETIs, not unlike hydrothermal vents that occupy only a small fraction of the ocean's floor. The ecological range of scaffolded collectives that are subsequently endogenised can be broader than the (possibly limited) range of environments that promote their emergence in the first place.

## Methods

The main stochastic model is a Markov jump process, with events detailed in Supplementary Note 1. The simplified ODE model, analysed with the methods of adaptive dynamics is described in Supplementary Note 2. Simulations and numerical analysis were implemented in Python 3, using the numpy[100] and scipy[101] libraries. Visual representations were produced using the matplotlib library[102] and the inkscape software. The code and output used to produce all figures in this manuscript are available (see the Code Availability section).

### Reporting summary
Further information on research design is available in the Nature Portfolio Reporting Summary linked to this article.

## Data availability
Data sharing not applicable to this article as no datasets were generated or analysed during the current study. Simulation outputs featured in the figures are available in Supplementary Code 1.

## Code availability

The code used in the simulations, analysis and figures that feature in this study is available in Zenodo with the identifier: https://doi.org/10.5281/zenodo.8335843 under the CC-BY License. Code and simulation output to reproduce Figs. 2, 3 and 5 as well as all the Supplementary Figs. are provided.

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

## Acknowledgements

The authors thank the participants of the workshop "Evolutionary Transitions in Individuality" held at the Max Plank Institute for Evolutionary Biology in May 2023. The authors also thank Sarah Pearce for professional editing of the manuscript. The authors (GD, PT, KH, PB) gratefully acknowledge the financial support of the John Templeton Foundation (#62220). The opinions expressed in this paper are those of the authors and not those of the John Templeton Foundation. PB and PT's research was also supported under the Australian Research Council's Discovery Projects funding scheme (Project Numbers FL170100160 & DE210100303).

## Author contributions

G.D., P.T., K.H. and P.B. conceived the study together. G.D. wrote the model and performed the analysis. G.D., P.T., K.H. and P.B. contributed to writing the manuscript. K.H. and P.B. contributed equally.

## Competing interests

The authors declare no competing interests.
