## [Peer Review File · Nature Communications]

Stability of Ecologically Scaffolded Traits During Evolutionary Transitions in IndividualityReviewers' Comments:

Reviewer #1:

Remarks to the Author:

This manuscript details the effect of static, externally imposed metapopulations and uses ESS analysis to evaluate the possible trajectories of individual particle traits such as dispersal versus reproduction. In changing environments of varying lengths, the particle traits evolve to values which are interpreted as collective level traits, thus producing an evolutionary transition of individuality, being coordination between cells at the collective level. As the environment changes back, this trait is constant, producing a collective rather than particles. While interesting, there are several major issues with the manuscript that warrant rejection.

The manuscript is framed by ecological scaffolding, which confuses the concepts discussed in the paper. In this manuscript, and elsewhere in the literature, ecological scaffolding is poorly described and vague. As flashy terminology is no substitute for clear and articulate discussion of the concepts at hand, the authors should instead frame their results in population dynamics and selection. "Ecological scaffolding" adds no value to the manuscript while severely reducing the clarity of the manuscript.

Secondly, I do not have the sufficient expertise to evaluate the correctness of the mathematical model, I am very skeptical of the reliance on specific initial conditions in their simulations. For example, the trait θ is interpreted to be "the error of the sensory system" and initial conditions are assumed to be completely non functional, $\theta=0.5$. There is a local maximum of fitness at $\theta=0.57$, which suggests that a more error prone sensor system is somehow a higher fitness? The interpretation of "the error of the sensory system" also appears to confuse the manuscript by defining both $\theta=0$ and $\theta=1$ throughout the same section as perfectly accurate. So which is it? If this initial value is instead $\theta=0.2$, and particles tend to prefer replication, then based on the fitness landscapes shown in Figure 4, wouldn't θ evolve towards 0 and point 4 and stay there? If the result of "endogenization" is so critically important on arbitrary initial conditions, what have we really learned? This worry is similar to the framework of R seasonality- if it is only replaced when a niche is empty, as opposed to some consistent or fluctuating increase, how is this applicable? Would the same result occur under less artificial conditions of R seasonality?

Furthermore, there is a critical jump in the interpretation of θ approaching 0, early in section D.2, this trait value is interpreted as "the sensory system is perfect", but after R changes a couple of times, the same value of θ is suddenly interpreted as "coordination between particles" and appears to be the main reason for claiming to show ETIs are occurring in their model. There is no justification for this switch, and the authors have not shown that any collective trait has evolved rather than particles operating to increase fitness. This is best displayed by the authors themselves! The discussion of Sup Fig 7 highlights how this result is a re-working of Beisner et al 2003 in a bit more detail with resource amount changing and some spatiality in figure 5. In this view, this work is a small increment on the work of Beisner et al. Alternatively, from the view of the evolution of multicellularity, this work is extremely conceptually similar to the modifier models of Michod et al (so many papers), just adding a component of dispersal, simply focusing on dispersal rather than genetic. In this view, this manuscript is also highly incremental.

Reviewer #2:

Remarks to the Author:

Review of "stability of ecologically scaffolded traits during evolutionary transitions in individuality"

In this study, the authors build on their previous proposal of ecological scaffolding as a mechanism for evolutionary transitions in individuality. With ecological scaffolding, the environment provides a meta-population structure which facilitates the evolution of higher-level individuals from lower-level

particles. In particular, they identify the problem of stabilizing the collective after the scaffold has been removed or left behind by the collective. To maintain the collective, the new traits it evolved have to remain stable in absence of the scaffold, something they call endogenisation. The authors develop two models to illustrate conditions where endogenisation does or does not occur.

In general, the models are simple but illustrate the point nicely, and I agree that scaffolding is potentially one mechanism by which ETIs can occur, so it is worth investigating. However, I have some major and minor concerns that I would like to see addressed before publication of this work.

Major points:

My main concern is that the focus on “collective-level traits” means that the actual maintenance of the collective itself is ignored, and that this has big consequences for the formulation of the models. It appears to me that the formal model, described in section C, describes a form of seascape, or stepping stone evolution, which has already been discussed in many biological contexts. The same principle may certainly apply to ETIs, but it needs to be placed in its proper context. The problem of the formal model being so general becomes clear with the model in section D, which is supposed to represent a model in which endogenisation is taking place for an ETI. However, adding the scaffold results in the evolution of single-particle organisms with good sensors. This sounds more like quorum-sensing, which does not necessarily strike me as a collective-level trait, belonging to a collective that has undergone an ETI. I think it’s not necessarily a bad thing to have a general model which applies to ETIs as well as other traits, but this would need to be discussed in the pertaining section and the discussion. I would also appreciate a clearer explanation of how the model represents an ETI, and why the discussed trait is collective. Just sensing seems insufficient.

As an example: the spatial model presented at the end of section D strongly resembles the setup of a recent model by Von der Dunk et al (2022, doi.org/10.1093/gbe/evac056), which does not concern itself with, or show the emergence of, ETIs. This is a more sophisticated model where genome structure and regulation of the cell cycle evolves in a spatial environment with increasingly nutrient-poor sites, in a very similar spatial structure to the model in section D. In a subset of cases they observe the evolution of a generalist strategy which evolves in poorer environments, and then back-invades the richer environments.

I think that given this similarity, this model should absolutely be cited and discussed in relation to the work presented in the paper, and especially in light of their definition of scaffolding for ETIs and endogenization.

I also think that the discussion of other, more recent mathematical/computational models on evolutionary transitions is lacking. Many of these computational models show how selectable traits emerge and change at ETIs – see for instance work on the evolution of multicellularity, by Sole and collaborators, De Monte and collaborators, and Colizzi and collaborators, as well as work on the origin of life by Hogeweg and collaborators, to just name a few.

Minor points:

- I think another weakness of the formal model in this paper is that it doesn’t seem to translate very well to traits that are emergent properties of collectives and therefore do not exist at the single-particle level. This type of trait seems particularly relevant for ETIs. Could the authors discuss whether this type of trait can be endogenised, and how their formal model would map?

- Could the authors include some discussion on how their formal and simple models and the traits under discussion map to actual ETIs, like endosymbiosis or evolution of multicellularity? I have a hard time picturing this.

- The particle trait in the model of section D is a sensor that distinguishes between 1 or more particles in the niche, which, helpfully, would also allow a cell with a decent sensor to be very close to the stated global maximum of particle production. Would this model also work with a more sliding

function, or where the reaction norm for the number of particles as well as the probability would evolve? That would seem more biologically plausible to me.

- I don't quite understand why in section D, the minor ESS is at $\theta=0.57$; could this be briefly explained?

- On page 8 in the final paragraph of the discussion, I feel I might have misunderstood the main point of the study so far. I understood that not all traits can be endogenised, as endogenisation requires a particular trait-fitness landscape. But the meta-population structure in the models discussed seemed fairly similar – unless the amount of resources available in each niche is considered part of the structure? Could you please clarify this there?

Small remarks:

- I understood from the description of the model that when a propagule is produced, there is no duplication of the particle (i.e., propagule=particle movement to another niche). However, figures 2a and 4a seem to suggest otherwise, or the figure is somewhat confusing (the particle seems to both become a propagule and remain in the niche).

- Could you mention what is the dashed line in the legend of figure 2B?

- On page 3, 3rd paragraph, Supplementary figure 1 does not match the description in the text of what we should see there.

- On page 3, 5th paragraph, Supplementary figure 6 is referenced, but only half of the figure (the left) applies. Could you label them A and B and reference accordingly? (although the second part is never referred to). Also, the sentence after is a bit confusing, since there is a dashed line in supplementary figure 6 but not in supplementary figure 3, so I thought at first the reference was wrong).

- In the first paragraph of the discussion, could you list the citations for the claim that "discretisation and the evolution of collective traits have been singled out as particular important aspects of individuality"?

- on page 6, the reference to the fleet herd of deer metaphor seems a bit mangled (cited as "fleet of deer").

I found the supplementary notes a bit difficult to parse, which can probably be remedied with just a few additional explanations. In general, the figure legends should be a bit more explanatory.

Note 1:

- Are C and E densities or numbers? Because the initial conditions suggest the latter, but the description says the former

- please relist the definitions of rho and tau.

- the legend of supp fig. 1 lists equilibrium E^* , but on the y-axis it says C^*

Note 2:

- a bit hard to follow for those unfamiliar with Markov branching processes

- the description of the first case, $k>1$, is a bit confusing: what do t and x reference?

- Do you mean Bertrand's ballot theorem, rather than the Ballot theorem, which I could not find?

- please define C_n , or mention that it is the Catalan number?

Note 4:

- in supplementary figure 5, please label the lines so it is clear which one is rho and which one tau.

Note 5:

- condition 4 is not entirely obvious to me. Can't θ_s be larger than θ_m , and still have the same effect? The basin of attraction can be symmetrical, or at least extend to the right.

Object: **Response to the review**

Dear Reviewers,

Thank you for reviewing our article “Stability of Ecologically Scaffolded Traits During Evolutionary Transitions in Individuality” (NCOMMS-23-38854). We have taken your remarks and comments into account and have significantly revised the manuscript. We believe that the new version is much improved due to your comments. Thank you!

In the following, we have detailed our response to your reviews and identified the changes we have made to the manuscript.

Reviewer 1:

[1.1] “The manuscript is framed by ecological scaffolding, which confuses the concepts discussed in the paper. In this manuscript, and elsewhere in the literature, ecological scaffolding is poorly described and vague. As flashy terminology is no substitute for clear and articulate discussion of the concepts at hand, the authors should instead frame their results in population dynamics and selection. “Ecological scaffolding” adds no value to the manuscript while severely reducing the clarity of the manuscript.”

Reviewer 1 considers the concept of ecological scaffolding “poorly discussed and vague” but does not provide any argument for this statement beyond personal preference. Ecological scaffolding is a relatively novel but well-established concept in the field of evolutionary transitions in individuality (ETIs). It has been the object of several publications in both experimental evolution (Hammerschmidt et al. 2014), theoretical biology (Black et al. 2020, Doucier et al. 2020), and philosophy of science (Griesemer and Shavit 2023, Neto et al. 2023, Neto and Meynel 2023, Veit 2021, Bourrat 2021). Ecological scaffolding is not just “flashy terminology”; rather, it is a concept with novel content.

The core idea of ecological scaffolding is that the structure of the meta-population alone can promote an ETI by discretising collectives of particles and allowing natural selection to act on them. This second level of selection arises from the externally imposed population structure. While we agree with the reviewer that the model is ultimately about population dynamics and selection,

this specific pattern of population dynamics and selection in the context of ETIs has been described in the literature as ecological scaffolding.

Changes to the manuscript: While we disagree with the reviewer’s taste in terminology, we share their concern for precision and clear definitions. Therefore, we have amended the paragraph that first describes ecological scaffolding (l. 47, from “Recently”) and we have made the differences compared to other approaches clearer in the introduction (l. 65) and throughout the manuscript (particularly in the discussion).

[1.2] “Secondly, I do not have the sufficient expertise to evaluate the correctness of the mathematical model, I am very skeptical of the reliance on specific initial conditions in their simulations.” [...] “If this initial value is instead $\theta = 0.2$, and particles tend to prefer replication, then based on the fitness landscapes shown in Figure 4, wouldn’t θ evolve towards 0 and point 4 and stay there? If the result of “endogenization” is so critically important on arbitrary initial conditions, what have we really learned? This worry is similar to the framework of R seasonality- if it is only replaced when a niche is empty, as opposed to some consistent or fluctuating increase, how is this applicable? Would the same result occur under less artificial conditions of R seasonality?”

We are describing a bistable system with alternate stable states that correspond to an ancestral and a derived state. In this context, it is trivial that if the eco-evolutionary system has initial conditions in the basin of attraction of the derived state, it will evolve naturally towards it. This is actually illustrated in Supplementary Figure 6. However, we consider that the system is initially either at the eco-evolutionary equilibrium or in the basin of attraction of the ancestral state and let it relax to equilibrium. Therefore, there are only two possible initial conditions: the two alternate stable states. The phenomenon we describe (i.e., the evolution of collective level properties) can thus only be observed if the initial conditions correspond to the ancestral stable state; otherwise, the phenomenon we want to explain is assumed to have already happened.

Changes to the manuscript: To make this clearer, we have made the hysteresis/alternative stable state interpretation of the model more prominent in the text (see comment [1.4]). We have also clarified the initial conditions of the model. (Section D2, l. 260)

The reviewer put forward an interesting point about R seasonality: what would happen if the patch renewal was not tied to the depletion of resources but was constant or fluctuating?

We have completed the analysis of the adaptive dynamics model with a constant patch flow k .

$$\begin{cases} \frac{dC}{dt} = CE\frac{\rho}{\tau} - C\frac{1}{\tau} \\ \frac{dE}{dt} = -CE\frac{\rho}{\tau} + k \end{cases} \quad (1)$$

The ecological equilibrium is $(\frac{1}{\rho}, k\tau)$. Thus, the invasion fitness becomes:

$$f(r, m) = \frac{1}{M} \frac{dM}{dt} = \frac{1}{\tau(m)} \left[\frac{\rho(m)}{\rho(r)} - 1 \right] \quad (2)$$

That is, the same as with coupled patch renewal described in the manuscript. We had considered that it would make the article needlessly complex to introduce a new parameter k .

Changes to the manuscript: We now mention the result of this computation in supplementary note 1 (l. 706).

[1.3] “For example, the trait θ is interpreted to be “the error of the sensory system” and initial conditions are assumed to be completely non functional, $\theta = 0.5$. There is a local maximum of fitness at $\theta=0.57$, which suggests that a more error prone sensor system is somehow a higher fitness? The interpretation of “the error of the sensory system” also appears to confuse the manuscript by defining both $\theta=0$ and $\theta=1$ throughout the same section as perfectly accurate. So which is it?” [...] “Furthermore, there is a critical jump in the interpretation of θ approaching 0, early in section D.2, this trait value is interpreted as “the sensory system is perfect”, but after R changes a couple of times, the same value of θ is suddenly interpreted as “coordination between particles” and appears to be the main reason for claiming to show ETIs are occurring in their model.

We agree with the reviewer that using two kinds of terminology (sensory system and coordination between cells) made the explanation of our model slightly more contrived. In the revised version, we have removed mention of the sensory system throughout and discuss only “coordination between cells.”

We have changed the definition of the trait compared to the original manuscript to $\theta := 1 - \theta$, so $\theta = 1$ is a “perfect coordination between particles,” $\theta = 0.5$ is “independent particles,” and $\theta = 0$ is “perfect anti-coordination between particles.” This change is cosmetic, but enables a more natural reading of the trait, as higher values translate to more coordination.

Regarding the “uncoordinated stable branch” value ($\theta = 0.57$ in the original manuscript), it is true that for large niches, the “uncoordinated” stable state is not exactly at 0.5 (as can be seen in Supplementary Figure 7), and a small quantity of anti-coordination is actually an evolutionarily stable strategy (ESS) that would evolve if the initial conditions were $\theta = 0.5$.

Changes to the manuscript: We have made the interpretation of θ clearer in the main text by rewriting Section D2 (l. 244). However, this does not change the result of the model, as highlighted in response to the comment [1.2]. Starting from independent particles, the population would reach the stable state of slightly anti-coordinated particles, and a change in the environment (i.e., ecological scaffolding) would be required for the population to switch to the coordinated stable state.

[1.4] There is no justification for this switch, and the authors have not shown that any collective trait has evolved rather than particles operating to increase fitness. This is best displayed by the authors themselves! The discussion of Sup Fig 7 highlights how this result is a re-working of Beisner et al 2003 in a bit more detail with resource amount changing and some spatiality in figure 5. In this view, this work is a small increment on the work of Beisner et al.”

We agree that Beisner et al. (2003), whom we cited in the caption of Sup Figure 7, described the concept of alternative stable states, and they exist in our model. However, alternative stable states are a very general phenomenon in dynamical systems that is featured in classic texts books (e.g. Murray 2001, p 8); it is also considered ubiquitous in biology, as reviewed by Noori in his book “Hysteresis Phenomena in Biology” (2013): “Hysteresis may occur in different spatiotemporal scales of consideration. From switches in protein-DNA interactions (Chatterjee et al. 2008), microscopic cellular signaling pathways with bistable molecular cascades (Angeli et al. 2004; Qiao 2007), cell division, differentiation, cancer onset and apoptosis (Sha et al. 2003; Eissing et al. 2004; Kim et al. 2007; Wilhelm 2009), protein folding (Andrews et al. 2013) and purinergic neuron-astrocyte interactions in the brain (Noori 2011) up to macroscopic biomechanics of cornea (Congdon et al. 2006) and lung deformations (Escolar and Escolar 2004), hysteresis phenomena are ubiquitous in biology.” Claiming that our model—because it uses this concept—is merely incremental is thus incorrect. Using the same reasoning, any work in evolutionary biology using the concept of natural selection would be incremental to Darwin and Wallace because they were the first to use the concept.

Further, one notable difference between our model and the concept discussed by Beisner et al., which was completely missed by the reviewer, is that Beisner et al.’s article discusses *ecological* alternative stable states (the state variable that has alternative stable states is population size, and the dynamics are ecological), while our model features *evolutionary* stable states (the state variable is the average trait in the population, and changes comes from successive invasions of mutant traits: the dynamics are evolutionary). Finally, Beisner’s article is not concerned with ETIs.

For these three reasons—(i) the ubiquity of hysteresis in biological systems, (ii) the evolutionary rather than ecological perspective, and (iii) the absence of the topic of ETIs—our work is not merely incremental to Beisner et al.’s.

Changes to the manuscript: We have changed the manuscript to make the connection with work on the phenomenon of hysteresis clearer in the introduction (l. 88), in section C (l. 203), and in the discussion (3rd part, l. 432). We now cite Noori (2011) in addition to Beisner et al.’s article.

[1.5] “Alternatively, from the view of the evolution of multicellularity, this work is extremely conceptually similar to the modifier models of Michod et al (so many papers), just adding a component

of dispersal, simply focusing on dispersal rather than genetic. In this view, this manuscript is also highly incremental.”

Our model only follows the “Modifier’s Model” strategy in the sense of Otto (2014)—that is, because it involves studying alleles that modify a feature of interest (here, migration rate) of individuals by explicitly modeling the evolutionary dynamics of the population (in our case, through adaptive dynamics) and determining “what evolves” (through invasion analysis). Michod’s Modifier Model, in contrast, asks whether a new allele is optimal from the point of view of the individual, lineage, or group. This approach is different and, we would argue, better because it does not presuppose optimality at either level. Further, it allows taking into account the effect of phenomena at the level of the lifespan of a particle or long-term events (e.g., patch-wide events of colonisation and collapse). We claim that these features are necessary for the adequate study of ETIs, and many models of ETIs operate following this strategy.

However, we disagree that our model is similar to Michod’s Modifier Model (2003, 2005, 2006a, 2006b, 2006c, 2007) because it features an explicit metapopulation structure with resources dynamics. As a consequence, the fertility-viability tradeoff in our model is not the consequence of an initial investment assumption (Michod 2006, JTB) that “filters up” at the collective level through the assumption that collective level traits are the average of particle traits, allowing the group-covariance effect (Michod 2006, PNAS). In our model, collective-level traits are a consequence of the population dynamics of cells: the fertility and lifespan of a collective is not the average of fertility and lifespan of the particles within it but, respectively, the expected number of propagules and of a collective seeded by one particle, and the expected time it takes for a patch seeded by one particle to become empty again.

We believe this approach has value because it relaxes an important assumption of collective behaviour (taking the expected value of a nonlinear dynamic rather than the average of a static trait) and places emphasis on how the metapopulation structure makes such a computation natural even without prior assortment mechanisms.

Changes to the manuscript: We have rewritten the third paragraph of the discussion (Discussion, 3rd paragraph, l. 384) to address the problem of “genuine” collective traits. We cite other models, such as Michod’s Modifier Model, in this paragraph (l. 402). We also have added the clarification that collective traits are not simple functions of particle traits in the presentation of the model (Discussion 3rd paragraph l. 402, Section B, l. 140)

Reviewer 2:

[2.1] “My main concern is that the focus on “collective-level traits” means that the actual maintenance of the collective itself is ignored, and that this has big consequences for the formulation of the models. It appears to me that the formal model, described in section C, describes a form of seascape, or stepping stone evolution, which has

already been discussed in many biological contexts. The same principle may certainly apply to ETIs, but it needs to be placed in its proper context.”

Dynamic Fitness Seascapes (Mustonen and Lassig, 2009) are fitness landscapes that change through time. This is the case in Figure E. However, our model goes further because the fitness landscape depends on time (in the sense that in Section D3, the scaffolding is time dependent) but also on the ecological stable state of the resident population since it is an adaptive dynamics model (Metz 1992, Geritz 1998).

Our model is a stepping stone model in the sense that dispersal is not possible towards any patch but only towards neighbouring patches (in contrast to Island models). Kimura (1953, 1982) introduced this in the context of the evolution of dispersal. However, the fact that dispersal is only possible towards empty patches distinguishes our model from a simple model of dispersal.

The “maintenance of collectives” is considered in our work through the co-evolutionary dynamics: values of θ that lead to populations with $\rho < 1$ are not viable and move inexorably towards extinction.

Changes to the manuscript: We now cite the proper Adaptive Dynamics article (Section B, l. 175) and mention the dynamic fitness seascapes in the Discussion (l. 350). We now mention in Section A of the results that the population of collective is not “maintained” if $\rho < 1$ and explain how this condition translates to particle-level traits. We have extensively reworked the second paragraph of the Discussion (Discussion 2nd paragraph, l. 327) to situate the model in the context of the literature. We now cite and contrast Levins’ meta-population model, as well as Kimura’s stepping stone model (Discussion 2nd paragraph, l. 377).

[2.2] “The problem of the formal model being so general becomes clear with the model in section D, which is supposed to represent a model in which endogenisation is taking place for an ETI. However, adding the scaffold results in the evolution of single-particle organisms with good sensors. This sounds more like quorum-sensing, which does not necessarily strike me as a collective-level trait, belonging to a collective that has undergone an ETI. I think it’s not necessarily a bad thing to have a general model which applies to ETIs as well as other traits, but this would need to be discussed in the pertaining section and the discussion. I would also appreciate a clearer explanation of how the model represents an ETI, and why the discussed trait is collective. Just sensing seems insufficient.”

Our model is a part of a more general research program concerning the role of an externally imposed population structure on ETIs. It is a model of ETIs to the extent that we try to explain the origin of a collective behaviour between particles (i.e., coordinated migration and reproduction at the level of the patch) through collective-level selection.

Our model is in line with a widespread view of ETIs with its essential steps: the assembly of groups (group formation) externally imposed by the scaffolding structure, the selection of collective-level properties through the selection of particle variants that turn out to maximise collective-level fertility (group transformation). We also point out that this selected behaviour is more than mere adaptation of particles to a structured environment due to the endogenisation phenomenon: the collective-level behaviour is stable even in the face of changes that would prevent its evolution in the first place.

Changes to the manuscript: We have substantially reworked the second paragraph of the Discussion to better situate the model in context. In particular, we give specific examples of how it could be adapted to treat different ETIs, such as the evolution of the cell, multicellular organisms, or eusocial groups (l. 361). We have dropped the interpretation of theta as a sensor mechanism throughout the manuscript to clarify the presentation (see point [1.3]) We have also reworked the third paragraph of the Discussion to make a better case about what constitutes a “genuine” collective trait (l 384).

[2.3] “As an example: the spatial model presented at the end of section D strongly resembles the setup of a recent model by Von der Dunk et al (2022, doi.org/10.1093/gbe/evac056), which does not concern itself with, or show the emergence of, ETIs. This is a more sophisticated model where genome structure and regulation of the cell cycle evolves in a spatial environment with increasingly nutrient-poor sites, in a very similar spatial structure to the model in section D. In a subset of cases they observe the evolution of a generalist strategy which evolves in poorer environments, and then back-invades the richer environments. I think that given this similarity, this model should absolutely be cited and discussed in relation to the work presented in the paper, and especially in light of their definition of scaffolding for ETIs and endogenization.”

The model by Von der Dunk et al. is an interesting model of genetic regulation in an environmental structure with a resource gradient. However, this is almost where the similarity ends, as the spatial structure of Von der Dunk et al. is a regular lattice with 8 neighbouring cells, thus approximating a continuous cell lawn. In our model of ecological scaffolding, however, the spatial structure is organised into resource patches (that are smaller or larger depending on the gradient), and all cells within a patch die of starvation when its resources are depleted. There is no diffusion of resources between patches, and no cells take over neighbouring cells during their growth. Resource renewal occurs only through patch renewal, and cell death occurs only through patch resource depletion.

The questions are also different, as pointed out by the reviewer. Von der Dunk et al. examine genetic regulation and genome size of individual cells rather than collective-level behaviour.

Changes to the manuscript: We have added the following sentence to the discussion: “*Note that the fact that a variant can evolve under poor resource*

conditions and back invade regions of a gradient where it was unable to evolve is not a phenomenon restricted to ETIs and ecologically scaffolded populations, Von der Dunk et al. (2022), for example, show that this can happen in a non-patch structured continuous spatial population.”

[2.4] “I also think that the discussion of other, more recent mathematical/computational models on evolutionary transitions is lacking. Many of these computational models show how selectable traits emerge and change at ETIs – see for instance work on the evolution of multicellularity, by Sole and collaborators, De Monte and collaborators, and Colizzi and collaborators, as well as work on the origin of life by Hogeweg and collaborators, to just name a few.”

This is a fair request.

Changes to the manuscript: We have extensively reworked the second paragraph of the discussion to better place the model in context. We now cite and discuss the following models: Colizzi et al, 2022 (l. 374), Vroomans et al., 2020 (l. 337), Libby and Ratcliff, 2016 (l. 423), Garcia et al. 2015, (l. 337), De Monte and Miele 2021, (l. 66), Hermsen, 2022 (l. 343), and Pichugin, 2017 (l. 333), Bonforti and Solé (l. 354).

[2.5] “I think another weakness of the formal model in this paper is that it doesn’t seem to translate very well to traits that are emergent properties of collectives and therefore do not exist at the single-particle level. This type of trait seems particularly relevant for ETIs. Could the authors discuss whether this type of trait can be endogenised, and how their formal model would map?”

There are two kinds of trait discussed in our manuscript: particle-level traits, such as the probability for a cell to reproduce or migrate to a new empty patch (θ , $p...$), and collective-level traits, such as the collective life-history traits of fertility (ρ) and lifespan (σ).

Collective-level traits are emergent from the particle-level traits and the nature of the scaffold. To be more precise, the value of the collective-level traits can be computed from the particle-level traits and the number of resources in a patch. These collective-level traits are consistent with a weak-emergence concept of collective properties: they can only be understood when considering the dynamics of multiple particles but are nothing more than the result of interactions of particles in a given environment.

Note that collective life history traits in this model are the result of simple but explicit lower-level dynamics and do not rely on a classic assumption in ETI models that collective-level traits are simply the average of particle level traits.

Changes to the manuscript: We have reworked the third paragraph of the Discussion (l. 400) to more clearly discuss what are the collective-level and particle-level traits in our manuscript. Also, we have added the clarification that collective traits are not simple functions of particle traits in the presentation of the model (Discussion 3rd paragraph l. 402, Section B, l. 140) as mentioned in the response to comment [1.5].

[2.6] “Could the authors include some discussion on how their formal and simple models and the traits under discussion map to actual ETIs, like endosymbiosis or evolution of multicellularity? I have a hard time picturing this.”

Changes to the manuscript: We have reworked the second paragraph of the Discussion to include how the model could be modified to treat different ETIs, such as the evolution of the cell, multicellular organisms, or eusocial groups (l. 361). We now discuss the specific case of endosymbiosis if the scaffold is not abiotic within the Discussion. We also discuss the fact that multicellularity might not require cell-cell attachment (Lyons 2015), so models about coordination are relevant (l. 354)

[2.7] “The particle trait in the model of section D is a sensor that distinguishes between 1 or more particles in the niche, which, helpfully, would also allow a cell with a decent sensor to be very close to the stated global maximum of particle production. Would this model also work with a more sliding function, or where the reaction norm for the number of particles as well as the probability would evolve? That would seem more biologically plausible to me.”

This is an interesting suggestion. The model would likely also work with an imperfect sensor or with an additional particle death term. We expect that it would change the optimal strategy, so the optimal coordination level would not be exactly full coordination but something slightly less drastic that would lead to larger population size within collectives (and thus a lower absolute collective fertility). We have not studied this version of the model because it would require a substantial change to the formal apparatus without qualitatively changing the overall message, as the hysteresis behaviour would stay. This could be a path to be followed in future research. In particular, it could be linked to known selection patterns for higher collective size.

[2.8] “I don’t quite understand why in section D, the minor ESS is at $\theta=0.57$; could this be briefly explained?”

As shown in Supplementary Figure 7, there are two ESS for large patches: one that is perfect coordination between particles ($\theta = 0$ in the original submission, now $\theta = 1$), and another that is close to independent particles ($\theta = 0.5$). However, the non-coordinated stable state is not always exactly at $\theta = 0.5$ but slightly more or less coordinated. This does not change the conclusion of the model, as we still have two alternative stable states with qualitatively different coordination behaviours between cells.

Changes to the manuscript: We have reworked the description of the model in Section D2 to clarify this point (as also mentioned in the response to comment [1.3]).

[2.9] “On page 8 in the final paragraph of the discussion, I feel I might have misunderstood the main point of the study so far. I understood that not all traits can be endogenised, as endogenisation requires a particular trait-fitness landscape. But the meta-population structure in the models discussed seemed fairly similar – unless the amount of resources available in each niche is considered part of the structure? Could you please clarify this there?”

The number of resources, to the extent that it constrains the carrying capacity of the niche, is a crucial (quantitative) part of the scaffolding structure.

Changes to the manuscript: To make this clearer, we have added “limited resources patches” in the paragraph first introducing ecological scaffolding in the Introduction.

[2.10] “I understood from the description of the model that when a propagule is produced, there is no duplication of the particle (i.e., propagule==particle movement to another niche). However, figures 2a and 4a seem to suggest otherwise, or the figure is somewhat confusing (the particle seems to both become a propagule and remain in the niche).”

This mistake has been corrected: the blue circle in the right side of the red arrow has been removed.

[2.11] “Could you mention what is the dashed line in the legend of figure 2B?”

This has been fixed. The dashed line corresponds to the optimal value of $(\theta^*, \rho(\theta^*))$ for all values of R between 2 and 100.

[2.12] “On page 3, 3rd paragraph, Supplementary figure 1 does not match the description in the text of what we should see there.”

This is now corrected: the main text now refers to the actual content of Supplementary Figure 1.

[2.13] “On page 3, 5th paragraph, Supplementary figure 6 is referenced, but only half of the figure (the left) applies. Could you label them A and B and reference accordingly? (although the second part is never referred to). Also, the sentence after is a bit confusing, since there is a dashed line in supplementary figure 6 but not in supplementary figure 3, so I thought at first the reference was wrong).”

This is now corrected: Supplementary Figure 6 now has labeled panels. We also edited the following sentence for clarity. It now reads: “Examples of stochastic trajectories are shown in Supplementary Figure 6a,b. The value of the ESS for each value of R is shown Figure 2b (dashed line) and Supplementary Figure 3.”

We added a reference to panels c and d of Supplementary Figure 6 in Section D3.

[2.14] “In the first paragraph of the discussion, could you list the citations for the claim that “discretisation and the evolution of collective traits have been singled out as particular important aspects of individuality”?”

These are the references for this claim: Godfrey-Smith (2008), Clarke (2013, 2014), Bourrat (2021 Fact...), Bourrat (2021 Transition...), Bourrat (2022). However, the new Discussion does not make this statement using this exact wording (although those work are still cited in 1).

[2.15] “Oon page 6, the reference to the fleet herd of deer metaphor seems a bit mangled (cited as “fleet of deer”).”

This is now corrected. We have also added the example when discussing the difference between a particle-level trait and a genuine collective-level trait at the beginning of the new Discussion.

[2.16] Note 1:

- Are C and E densities or numbers? Because the initial conditions suggest the latter, but the description says the former
- please relist the definitions of rho and tau.
- the legend of supp fig. 1 lists equilibrium E^* , but on the y-axis it says C^*

C and E are densities. We have clarified this point. We have recalled the definition of τ and ρ . The legend of the figure has been corrected.

[2.17] Note 2:

- a bit hard to follow for those unfamiliar with Markov branching processes
- the description of the first case, $k > 1$, is a bit confusing: what do t and x reference?
- Do you mean Bertrand’s ballot theorem, rather than the Ballot theorem, which I could not find?
- please define C_n , or mention that it is the Catalan number?

It is indeed the Bertrand’s ballot theorem. The C was the binomial coefficient; we now use Ettingshausen’s $\binom{n}{k}$ notation rather than the less widespread C_n^k notation used previously. We have clarified the demonstration.

[2.18] Note 4: - in supplementary figure 5, please label the lines so it is clear which one is rho and which one tau.

This is now done.

[2.19] Note 5: condition 4 is not entirely obvious to me. Can't θ_s be larger than θ_m , and still have the same effect? The basin of attraction can be symmetrical, or at least extend to the right.

That is correct. We have edited condition 4.

Sincerely,

Guilhem Doucier, Katrin Hammerschmidt, Peter Takacs and Pierrick Bourrat.

Reviewers' Comments:

Reviewer #1:

Remarks to the Author:

The authors have not addressed my largest concern, the use of terminology to inflate the importance of their work. They readily agree the concepts at hand are "ultimately about population dynamics and selection", yet oddly are unwilling to frame their research as the effects of population dynamics and selection. While important and interesting, their usage of "ecological scaffolding" to disguise the plain novelty of their work is unjustified.

The authors have added "collective traits evolve as a consequence of a change in the environment (i.e., the scaffolding)". Traits, no matter what level, evolving as a consequence of the environment is "selection". The authors are remiss to not use the existing concepts and frameworks, instead relying on artificially created concepts (ecological scaffolding) to discuss already well understood concepts.

The same is true of "endogenization", which is "figuratively depicted as the population retaining a memory of the scaffold". The rest of the community calls this concept "historical contingency", yet the present authors insist upon inventing terminology rather than clearly and coherently discuss the novelty of their work, the effect of historical contingency on unique population structure under fluctuating selection pressures.

To their credit, in their revisions, the authors clarify what novel contributions they make compared to previous work. This "model is different in that the migration of particles towards patches that are already occupied is considered negligible." At the same time, they allow overlapping collective generations and use a more complex spatial structure. I am concerned that these modifications are relatively incremental, though again, their refusal to use commonly accepted conceptual frameworks like historical contingency, selection, and population structure makes assessing this novelty challenging.

Also to their credit, the authors have addressed my other concerns sufficiently.

Reviewer #2:

Remarks to the Author:

The authors have addressed my concerns and I find that the manuscript is clearer and easier to read than before.

I agree with the authors that the sensing in the model in section D may indeed be interpreted as a collective trait and it now comes out of the text more clearly.

I do think that the models apply to both ETI and non-ETI trait evolution, but they may still serve as clarifying examples of the concept of scaffolding.

I have a few minor corrections:

- in line 113 and 138, the authors refer to collectives; I assume that collective = the set of particles within a patch, be that 1 or more? Could you add a single sentence specifying this? On first rereading, I thought patch and collective were completely the same thing.

- line 135: "cell" should probably be "particle"

- section A/B: could you specify in the main text that the graph of the meta-population is fully connected?

- line 902: The description of supplementary figure 4 was a bit confusing to me: can these be

considered trajectories (a term I associate with temporal dynamics)? The figure itself is clear, depicting the steady state occupancy of niches given a fixed trait value p and resource value R .

Reviewer #3:

Remarks to the Author:

In this thought provoking article, Doucier and collaborators address an important question in the context of Evolutionary Transitions in Individuality: How are collective traits maintained, once the environmental conditions in which they evolved change? This question is important and timely in the field of ETI's, an example of which is the evolution of multicellularity. During an ETI, we do expect that collective-level traits (e.g. cell-cell communication) once evolved become endogenised in the nascent higher-level individual, and once the transition is complete, these traits are maintained through evolution. But through which mechanism this may happen is not clear. This work extends a previous line of research to make a strong case for how such endogenisation can happen. The authors clarify the conditions for the evolutionary maintenance of collective traits, once these have evolved in "scaffolding" environments.

These conditions essentially are that the trait under consideration is evolutionarily bistable under some environmental conditions, called "non-scaffolding", and have a unique evolutionary stable steady state under other environment conditions, called "scaffolding". When these conditions are met, the trait evolved under scaffolded conditions will persist (i.e. it is "endogenised") when the scaffold is removed. This result gives us an expectation of when a trait may or may not be maintained by the collective. After this, they give an example of one such trait evolving first in a temporally and then spatially structured environment. In essence, the proposed solution to the problem of endogenisation requires us to consider both environment structure and the specifics of the traits that is being endogenised.

I think the results are original, and the conclusions valid. Moreover, I feel this paper moves the discussion from abstract arguments on fitness and division of labours, to concrete traits, functions and their effects on the different levels of organisation. I expect this paper will inspire further discussion on the topic. I see that my review comes after a previous round of review. The other two reviewers already covered many points about methodology, and these have been addressed by the authors. I think the paper is pretty much ready for publication, and I only have a couple of small remarks that I would like the authors to address.

1. Line 145: The word "Additionally" is a bit oddly placed at the beginning of the paragraph.

2. Line 225 (Section D). Can the authors give an example of a trait that looks like the one that is modelled here? This would help make the model more concrete.

3.1 Unless I misunderstood it, one thing that does not happen in the model is an ETI. That is, after evolution there is no "new" higher-level individual. If I am wrong, could you please make clear what is the higher-level individual evolved here, and otherwise, how can the model be extended to achieve an ETI? I think this could make for a good discussion point.

3.2 Related to this, after the collective trait is endogenised, the "collectives" are made of a single particle. Can you please discuss how this can be changed? E.g. looking at fig. 4C, it seems to me that this depends on the fitness function having a maximum for $\theta=1$. This is no fatal flaw of the model, but can you please discuss how to change this?

4. Line 481: I was surprised to suddenly see a lot of details about one specific hypothesis for the origin of life. I would like to note that almost all hypotheses of the origin of life consider a special location (hydrothermal vents, intertidal zones, liquid bubbles in the atmosphere, underwater streams...) and a special chemistry that could only occur there. So this choice in particular seems odd. Besides, biological evolution is not really considered in any of these hypotheses, and therefore there is

no guarantee that those dynamical systems bear any similarity with the one developed here. If the point is to draw a connection with ETI's, there is a lot of work on (Emergent) multilevel evolution from the Hogeweg group and the Szathmary group, showing many examples of the evolutionary interplay between collective traits (metabolic or cooperative interactions) and spatial structure (including self-structuring and compartmentalisation).

Kind regards,
Enrico Sandro Colizzi

We would like to thank all three reviewers for their comments on our manuscript. We have addressed all their comments (our changes are highlighted in the second pdf file). We believe that the new version is much improved, due to these comments.

In the following, we provide detailed responses to each comment and refer to the corresponding changes we have made to the manuscript.

Reviewer 1

[1.1] *“The authors have not addressed my largest concern, the use of terminology to inflate the importance of their work. They readily agree the concepts at hand are “ultimately about population dynamics and selection”, yet oddly are unwilling to frame their research as the effects of population dynamics and selection. While important and interesting, their usage of “ecological scaffolding” to disguise the plain novelty of their work is unjustified.*

The authors have added “collective traits evolve as a consequence of a change in the environment (i.e., the scaffolding)”. Traits, no matter what level, evolving as a consequence of the environment is “selection”.

The authors are remiss to not use the existing concepts and frameworks, instead relying on artificially created concepts (ecological scaffolding) to discuss already well understood concepts.

The same is true of “endogenization”, which is “figuratively depicted as the population retaining a memory of the scaffold”. The rest of the community calls this concept “historical contingency”, yet the present authors insist upon inventing terminology rather than clearly and coherently discuss the novelty of their work, the effect of historical contingency on unique population structure under fluctuating selection pressures.

To their credit, in their revisions, the authors clarify what novel contributions they make compared to previous work. This “model is different in that the migration of particles towards patches that are already occupied is considered negligible.” At the same time,

they allow overlapping collective generations and use a more complex spatial structure. I am concerned that these modifications are relatively incremental, though again, their refusal to use commonly accepted conceptual frameworks like historical contingency, selection, and population structure makes assessing this novelty challenging”

We share the reviewer’s concern about the proliferation of jargon and the necessity of clear terminology. However, we strongly deny the allegation that our use of “ecological scaffolding” is an attempt to “disguise the plain novelty of [our] work.” We believe that ecological scaffolding—as a very specific scenario involving environmental changes, meta-population structure, population dynamics, and natural selection—is definite enough to warrant its own term. It is not uncommon in the field to use terminology that pertains to a specific evolutionary scenario with a uniquely circumscribed set of environmental conditions or historical contingencies. Consider the concept of evolutionary rescue coined in 1995 by Gomulkiewicz & Holt. It is used in coupled eco-evolutionary models to identify the conditions under which a population subject to an abrupt environmental change that significantly lowers its density can be “rescued” by evolution. That new concept has been fruitfully applied in conservation biology and evolutionary medicine (Bell, 2019), despite being only shorthand for a more specific eco-evolutionary arrangement (the sudden stress and change in selection) and corresponding research question (Has the population been rescued?). We consider that by using “ecological scaffolding,” we are making a similar type of move, one that profitably galvanises a community’s research around the arrangement of a meta-population and would-be consequences for ETIs.

Using the generic “historical contingency,” as suggested by the reviewer, is far too vague to be useful in the context of the manuscript. Our objective is more precise—we want to know how the interplay of a *very specific kind* of historical contingency (the meta-population structure with limited mixing, the uncrossable fitness valley) and natural selection can produce a well-defined outcome (the emergence of collective behaviors). The term “historical contingency” does not adequately reflect the fact that the system is apparently “‘retaining a memory’ of the scaffold” (note that this expression was introduced with scare quotes in the manuscript). Further, the phenomenon of hysteresis that we mention in the very next sentence (l.203) is widely accepted and understood. The introduction of hysteresis at this point in the discussion is clearly designed to “cache out” the metaphorical expression that the reviewer identifies as worrisome.

Additionally, as stated in our previous letter, ecological scaffolding is not a concept that we have “artificially created” for the present manuscript. Rather, it has been the object of several publications across fields in experimental evolution (Hammerschmidt et al. 2014), theoretical biology (Black et al. 2020, Doucier et al. 2020), and philosophy of science (Griesemer and Shavit 2023, Neto et al. 2023, Neto and Meynel 2023, Veit 2021, Bourrat 2021). While the reviewer may not agree with this body of work, they do not provide substantive reasons for disagreement in the review. It is not, as the reviewer seems to imply, a mere matter of terminological or conceptual preference. Ecological scaffolding and

endogenisation together provide a more comprehensive (non-question-begging) framework of explanation. The reviewer apparently concedes as much when they note that “the authors clarify what novel contributions they make compared to previous work.”

Despite disagreeing with several specific aspects of the comment, we genuinely appreciate the reviewer’s exigence and have now revised the manuscript to make the concept of ecological scaffolding as clear and accessible as possible. In particular, we have now made clearer that ecological scaffolding is an evolutionary scenario.

Changes to the manuscript:

- The sentence “if collective traits evolve as a consequence of a change in the environment (i.e., the scaffolding)” has been replaced l. 455 by “if collective traits are **selected** as a consequence of a change in the environment (i.e., the scaffolding)”
- The sentence “Endogenisation could be figuratively depicted as the population ‘retaining a ‘memory’ of the scaffold” has been replaced by “Endogenisation could be figuratively depicted as the population ‘retaining a memory’ of its scaffold. However, the sense of retention thereby implied must be understood as stronger than that which might be associated with merely encountering and evolving in the wake of some historically contingent set of circumstances.” (l. 204)
- We no longer discuss an ecological scaffolding “framework,” but rather an ecological scaffolding “scenario,” to put the emphasis on the fact that it pertains to a specific eco-evolutionary setting.
- In the introduction, we have added (l. 49): “Another benefit is that ecological scaffolding provides a unifying scenario for a range of research questions in the study of ETIs. It does not constitute a new evolutionary principle but rather a novel way of approaching ETIs that takes into account meta-populations and environmental gradients from evolutionary ecology (Levins 1969), self-renewing networks of patches from epidemiology (Kiss 2017, the effect of population structure and assortment on the evolution of cooperation studied from multi-level selection and game theory (Nowak 1992, Hogeweg 2002), and feedback between evolutionary and ecological processes from adaptive dynamics (Geritz 1998). This is in marked contrast to prevailing explanations of ETIs (Michod 2005, Bourke 2011, Ratcliff 2012, 2017, Miele 2021), which have traditionally attributed a less prominent role to the combination and interaction of these factors.
- We have made other small changes, such as replacing “This phenomenon is referred to as endogenisation” by “Proponents of ecological scaffolding have dubbed this phenomenon endogenisation” (l. 69).

[1.2] *Also to their credit, the authors have addressed my other concerns sufficiently.*

Again, we thank the reviewer for their fruitful criticisms.

Reviewer 2

[2.1] *“The authors have addressed my concerns and I find that the manuscript is clearer and easier to read than before. I agree with the authors that the sensing in the model in section D may indeed be interpreted as a collective trait and it now comes out of the text more clearly. I do think that the models apply to both ETI and non-ETI trait evolution, but they may still serve as clarifying examples of the concept of scaffolding.”*

We would like to thank the reviewer for this helpful comment. As the reviewer notes, the model might well apply to non-ETI trait evolution. Determining whether it in fact does so falls beyond the scope of the present manuscript. Yet, the possibility of a more general interpretation of the ecological scaffolding model that would include applicability to non-ETI trait evolution is broached at the end of the discussion (paragraph starting at the l. 481, “The ecological scaffolding scenario...”). We there discuss the application of the theory to artificial selection of communities, as undertaken in synthetic biology. Such consortia are not typically considered ETIs per se.

Changes to the manuscript: We have added the following sentence to the discussion: “Although beyond the scope of the present article, this raises the intriguing prospect that the ecological scaffolding scenario may also apply more generally to non-ETI trait evolution” (l. 495).

[2.2] *“in line 113 and 138, the authors refer to collectives; I assume that collective = the set of particles within a patch, be that 1 or more? Could you add a single sentence specifying this? On first rereading, I thought patch and collective were completely the same thing.”*

Yes, this is correct. Patch and collective are closely intertwined because each collective belongs to a patch and each patch can harbour a collective. However, the resources in a patch can be replenished when the collective (i.e., the last particle) dies. This latter fact helps distinguish the two notions (patch vs. collective).

Change to the manuscript: We have added the following sentence to the beginning of the section describing the model: “Each patch can harbour a set of (one or many) particles called a collective.” (l. 100)

[2.2] *“line 135: “cell” should probably be “particle”*

We thank the reviewer for alerting us to this and have corrected it.

[2.3] section A/B: could you specify in the main text that the graph of the meta-population is fully connected

This is now clarified.

Change in the manuscript: In section A, the sentence “As this stochastic process is a continuous-time Markov jump process” has been replaced by “Provided that the graph encoding the adjacency of patches is complete (i.e., that particles can migrate from any patch to any other empty patch), the process can be approximated by a simple set of two ordinary differential equations (ODEs).” The approximation holds for other kinds of graphs but is most accurate for complete graphs. In section B, we have changed “in the case of fully connected patches” to “we know from the analysis of the ODE system that, in the case of fully connected patches, the population is only viable if $\rho(\theta) > 1$ [...]” and changed “meta-population parameters (here, the number of resources in a patch)” (l. 181) to “(Here, the meta-population population is fully connected, the only parameter is the number of resources in a patch).”

[2.4] line 902: *The description of supplementary figure 4 was a bit confusing to me: can these be considered trajectories (a term I associate with temporal dynamics)? The figure itself is clear, depicting the steady state occupancy of niches given a fixed trait value p and resource value R .*

We agree that the wording in the supplementary text was not entirely clear. However, the figure does not show “trajectories.” The points show “the occupancy at steady state in the stochastic simulations,” which should be compared against the meanfield approximation (the line).

Changes to the manuscript: This sentence has been rewritten to be clearer and does not mention trajectories anymore (l. 903).

Reviewer 3

[3.1] *“In this thought provoking article, Doucier and collaborators address an important question in the context of Evolutionary Transitions in Individuality: How are collective traits maintained, once the environmental conditions in which they evolved change? This question is important and timely in the field of ETI’s, an example of which is the evolution of multicellularity. During an ETI, we do expect that collective-level traits (e.g. cell-cell communication) once evolved become endogenised in the nascent higher-level individual, and once the transition is complete, these traits are maintained through evolution. But through which mechanism this may happen is not clear. This work extends a previous line of research to make a strong case for how such endogenisation can happen. The authors clarify the conditions for the evolutionary maintenance of collective traits, once these have evolved in “scaffolding” environments.*

These conditions essentially are that the trait under consideration is evolutionarily bistable under some environmental conditions, called “non-scaffolding”, and have a unique evolutionary stable steady state under other environment conditions, called “scaffolding”. When these conditions are met, the trait evolved under scaffolded conditions will persist (i.e. it is “endogenised”) when the scaffold is removed. This result gives us an expectation of when a trait may or may not be maintained by the collective. After this, they give an example of one such trait evolving first in a temporally and then spatially structured environment. In essence, the proposed solution to the problem of endogenisation requires us to consider both environment structure and the specifics of the traits that is being endogenised.

I think the results are original, and the conclusions valid. Moreover, I feel this paper moves the discussion from abstract arguments on fitness and division of labours, to concrete traits, functions and their effects on the different levels of organisation. I expect this paper will inspire further discussion on the topic. I see that my review comes after a previous round of review. The other two reviewers already covered many points about methodology, and these have been addressed by the authors. I think the paper is pretty much ready for publication, and I only have a couple of small remarks that I would like the authors to address.”

We would like to thank the reviewer for their careful reading of our manuscript and their comments.

[3.2] Line 145: The word “Additionally” is a bit oddly placed at the beginning of the paragraph.

Thank you for pointing this out.

Change to the manuscript: The sentence now reads: “The population is not viable for the two extreme trait values, $\theta = 0$ and $\theta = 1$.”

[3.3] Line 225 (Section D). Can the authors give an example of a trait that looks like the one that is modelled here? This would help make the model more concrete.

Change to the manuscript: We have added a passage at the beginning of section D1 that gives a biological example of such a trait for cells and for self-replicating molecules: “this trait model accordingly provides a simple way of depicting the effect of a control mechanism (receptors and regulation), one which has the capacity to change the probability for a given particle to disperse or duplicate. If, for instance, the particles in question are cells, this could represent the effect of direct cell-cell signaling or that of a quorum sensing molecule that results in activating or repressing physiological pathways. If, in contrast, the in question are self-replicating molecules, non-covalent interactions could lead to change the probability of dispersal depending on the density.”

[3.3a] Unless I misunderstood it, one thing that does not happen in the model is an ETI. That is, after evolution there is no "new" higher-level individual. If I am wrong, could you please make clear what is the higher-level individual evolved here, and otherwise, how can the model be extended to achieve an ETI? I think this could make for a good discussion point

One of the proposals in the manuscript is that endogenisation represents an important criterion of individuality at the collective level. We propose this criterion to explicitly counter the claim that a collective trait that is not fully determined by the environment can serve as a criterion of individuality. Thus, when endogenisation occurs (end of the evolutionary transition) and the environment is reverted to a non-scaffolding condition (temporally or spatially), collectives score higher on individuality than when it does not.

Changes to the manuscript: We have edited the discussion to streamline the discussion of this point: "The evolutionary trajectory presented in this manuscript features an increase in individuality at the collective level, following the foregoing distinction between a collective-level byproduct and a genuine collective trait. Two situations are considered for comparison. In the first case (intrinsic particle dispersal-duplication ratio), the probability of duplicating or migrating is independent of external cues to the particle; a collection of particles only displays an aggregation of identical behaviour. In the second case (density-dependent dispersal-duplication ratio), this probability depends on the coordination between cells. Their behaviour is not reduced directly to the average over individual behaviour, as it would be if they were alone. Thus, the evolutionary dynamics described here begin with an ancestral state containing uncoordinated particles on a patch (a mere collective-level byproduct) that is replaced by a derived state where particle behaviour is coordinated and collectives are composed of coordinated particles. This derived state resides further along the evolutionary trajectory towards collective-level individuality."

[3.3b] Related to this, after the collective trait is endogenised, the "collectives" are made of a single particle. Can you please discuss how this can be changed? E.g. looking at fig. 4C, it seems to me that this depends on the fitness function having a maximum for $\theta=1$. This is no fatal flaw of the model, but can you please discuss how to change this

Thanks for this comment and pressing us to discuss this point.

First, a technical remark. After the trait is endogenised, the average collective size is actually strictly higher than 1 and around 1.5. Indeed, if we observe any collective at $\theta = 1$, either it is composed of two particles (and one of them will disperse soon) or it is composed of a single particle (and it will reproduce). Therefore, on average, the number of particles is higher than 1.

Responding more directly to the point of the reviewer, $\theta = 1$ being an endpoint is contingent on the fact that $\theta = 1$ is a fitness maximum. There is also a more intuitive explanation for this point: there is no advantage in

having more than one or two particles at any point in the patch. In a sense, the optimal “soma” size for this model is 1.5 particles, and any extra particle does not improve collective reproduction.

The decision to exclude particle death from the model was made on the grounds of simplicity alone. In particular, it permits a detailed study of the stochastic system (Supplementary section B) and provides analytical results. When this simplifying assumption is introduced, only two kinds of changes are possible: adding a particle by reproduction or removing a particle by dispersal. It follows that the expected value can be analytically described with relative ease.

Small changes to the model can be introduced to recover higher particle sizes: for instance, having probability change smoothly from value to value rather than following a step function. However, this complexifies the presentation without significantly changing the results, so we only describe this in a new Supplementary Note section. Additional investigations prompted by the reviewer’s comment led us to the discovery that the hysteresis effect may disappear for larger collective sizes. A detailed analysis of this point is beyond the scope of this manuscript, but we discuss it in the new Supplementary Note and offer the conjecture that the phenomenon may be more restrictive for large collective sizes.

Change to the manuscript: We have added a Supplementary Note explaining in detail why the collective sizes are small and how small alterations can be made to the model to change this (Supplementary Note 6 (l. 931), as well as Supplementary Figures 8 and 9). We have reworked the relevant part of the manuscript (l. 248) and refer to this Supplementary Note.

[3.4] Line 481: I was surprised to suddenly see a lot of details about one specific hypothesis for the origin of life. I would like to note that almost all hypotheses of the origin of life consider a special location (hydrothermal vents, intertidal zones, liquid bubbles in the atmosphere, underwater streams...) and a special chemistry that could only occur there. So this choice in particular seems odd. Besides, biological evolution is not really considered in any of these hypotheses, and therefore there is no guarantee that those dynamical systems bear any similarity with the one developed here. If the point is to draw a connection with ETI’s, there is a lot of work on (Emergent) multilevel evolution from the Hogeweg group and the Szathmari group, showing many examples of the evolutionary interplay between collective traits (metabolic or cooperative interactions) and spatial structure (including self-structuring and compartmentalisation)

First, abiogenesis is considered the first ETI (see e.g., Herron 2021) and should be treated as such. So, yes, taking this example is a way to draw a connection to ETIs. Following the Goldilocks zone hypothesis, we suppose precisely that scaffolding and endogenisation would occur in special locations that fulfill special conditions that depend on the dynamics of the systems. However,

we think that the dynamical systems described in this manuscript (duplication, colonisation density effects, hysteresis effects) are abstract enough that the general structure of the problem would be similar for the origin of life. In other words, the “special” locations that the reviewer mentioned (hydrothermal vents, intertidal zones, liquid bubbles in the atmosphere, underwater streams...), with their unique effect on the meta-population structure, could very well correspond to the “Goldilocks zone” we have theorised. We have decided to focus on hydrothermal vents in the manuscript, as this is a well-thought-through scenario.

Finally, thank you for drawing our attention to other ETI models that also feature spatial structure, such as those from the Hogeweg and Szathmary groups. We certainly agree that spatial structure is important. We also recognise the parallels between scaffolding and compartmentalised models.

Change to the manuscript: First, we now make clearer the connection between Goldilocks zones and “privileged area of the environments” such as hydrothermal vents and others (l 507.). Second, we now cite the work of Hogeweg (Takeuchi and Hogeweg, 2009, l. 374) and Szathmary (2023, l. 389) on multi-level selection.

Yours sincerely,

Guilhem Doulcier, on behalf of the authors.

Reviewers' Comments:

Reviewer #1:

Remarks to the Author:

The authors continue to invest in artificial terminology, and unsurprisingly they cite several of their own papers to validate this use of needlessly created terminology. While the authors may not agree with my reviews, they do not provide substantive reasons for the terminology in the manuscript. As the authors continue to fail to provide substantive demonstration of the "scenario" of ecological scaffolding, I must recommend rejection of this article.

Reviewer #3:

Remarks to the Author:

The authors have answered all my questions satisfactorily.

As previously mentioned, I think this manuscript represents a significant step forward in understanding Evolutionary Transitions in Individuality. Congratulations!

Very minor comment: Supp Fig 9, please label the subplots with a,b,c.

Object:**Revision of manuscript**

We would like to thank the reviewers for their work and insights. We have replied to reviewer 1 comments in our previous letter and acknowledge that we have not convinced them despite our best effort.

We thank Reviewer 2 for their comments, and have labelled Supplementary Fig. 9's subplots with a,b,c as required.